# Vascular control of the $CO_2/H^+$-dependent drive to breathe

**Colin M Cleary[1], Thiago S Moreira[2], Ana C Takakura[3], Mark T Nelson[4,5], Thomas A Longden[6], Daniel K Mulkey[1]\***

[1]Department of Physiology and Neurobiology, University of Connecticut, Storrs, United States; [2]Department of Physiology and Biophysics, University of São Paulo, São Paulo, Brazil; [3]Department of Pharmacology, University of São Paulo, São Paulo, Brazil; [4]Department of Pharmacology, University of Vermont, Burlington, United States; [5]Institute of Cardiovascular Sciences, Manchester, United Kingdom; [6]Department of Physiology, University of Maryland, Baltimore, United States

**Abstract** Respiratory chemoreceptors regulate breathing in response to changes in tissue $CO_2/H^+$. Blood flow is a fundamental determinant of tissue $CO_2/H^+$, yet little is known regarding how regulation of vascular tone in chemoreceptor regions contributes to respiratory behavior. Previously, we showed in rat that $CO_2/H^+$-vasoconstriction in the retrotrapezoid nucleus (RTN) supports chemoreception by a purinergic-dependent mechanism (Hawkins et al., 2017). Here, we show in mice that $CO_2/H^+$ dilates arterioles in other chemoreceptor regions, thus demonstrating $CO_2/H^+$ vascular reactivity in the RTN is unique. We also identify $P2Y_2$ receptors in RTN smooth muscle cells as the substrate responsible for this response. Specifically, pharmacological blockade or genetic deletion of $P2Y_2$ from smooth muscle cells blunted the ventilatory response to $CO_2$, and re-expression of $P2Y_2$ receptors only in RTN smooth muscle cells fully rescued the $CO_2/H^+$ chemoreflex. These results identify $P2Y_2$ receptors in RTN smooth muscle cells as requisite determinants of respiratory chemoreception.

**\*For correspondence:**
daniel.mulkey@uconn.edu

## Introduction

Blood flow in the brain is tightly coupled to neural activity. Typically, an increase in neural activity triggers vasodilation to increase delivery of nutrients and clear metabolic waste (*Chow et al., 2020*; *Iadecola, 2017*). Blood flow may also directly regulate neural activity and contribute to behavior; perhaps, the most compelling evidence supporting this involves vasoregulation by $CO_2/H^+$. In most cases, $CO_2/H^+$ function as potent vasodilators to provide support for metabolic activity and to maintain brain pH within the narrow range that is conducive to life (*Ainslie and Duffin, 2009*). However, $CO_2/H^+$ has the opposite effect on vascular tone in a brainstem region known as the retrotrapezoid nucleus (RTN) (*Hawkins et al., 2017*; *Wenzel et al., 2020*), where $CO_2/H^+$ is not just metabolic waste to be removed but also functions as a primary stimulus for breathing (*Guyenet et al., 2019*). Neurons (*Guyenet et al., 2019*; *Kumar et al., 2015*) and astrocytes (*Gourine et al., 2010*) in this region are specialized to regulate breathing in response to changes in tissue $CO_2/H^+$ (i.e. function as respiratory chemoreceptors) and so vasoconstriction may augment this function by preventing $CO_2/H^+$ washout and maintaining the stimulus to respiratory chemoreceptors. However, it is not known whether $CO_2/H^+$ induced vasoconstriction is unique to the RTN or also contributes to chemotransduction in other chemosensitive brain regions. Also, mechanisms underlying this specialized $CO_2/H^+$ regulation of vascular tone in the RTN are largely unknown. Previous work showed that RTN chemoreception involves $CO_2/H^+$-evoked ATP-purinergic signaling most likely from astrocytes (*Gourine et al., 2010*; *Wenker et al., 2012*); however, downstream cellular and molecular targets of this signal are not known.

Here, we show that $CO_2/H^+$ vascular reactivity in the RTN is unique compared to other chemoreceptor regions. Our evidence also suggests that $CO_2/H^+$ constriction in the RTN enhances chemoreceptor activity and respiratory output, while simultaneous dilation in other chemoreceptor regions may serve to limit chemoreceptor activity and stabilize breathing. We also show that endothelial cells and vascular smooth muscle cells in the RTN have a unique purinergic (P2) receptor expression profile compared to other levels of the respiratory circuit that is consistent with $P2Y_2$-dependent vasoconstriction. Further, at the functional level, we show pharmacologically and genetically using $P2Y_2$ conditional knockout ($P2Y_2$ cKO) mice and RTN smooth muscle-specific $P2Y_2$ re-expression that $P2Y_2$ receptors on vascular smooth muscle cells in the RTN are required for $CO_2/H^+$-dependent modulation of breathing.

## Results

### $P2Y_2$ receptors are differentially expressed by vascular smooth muscle cells in the RTN

Purinergic signaling is required for $CO_2/H^+$-induced constriction of RTN arterioles (*Hawkins et al., 2017*). Therefore, as a first step toward identifying the cellular and molecular basis of this response, we determined the repertoire of P2 receptors expressed by endothelial and smooth muscle cells associated with arterioles in the RTN. We also characterize the P2 receptor expression profile of vascular cells in the caudal aspect of the nucleus of the solitary tract (cNTS) and raphe obscurus (ROb), two putative chemoreceptor regions where P2 receptor signaling does not contribute to respiratory output (*Sobrinho et al., 2014*). For this experiment, we used Cre-dependent smooth muscle (*Myh11*[Cre/eGFP]) and endothelial cell (*Tek*[Cre]::TdTomato) reporter lines and performed fluorescence activated cell sorting to obtain enriched populations of endothelial or smooth muscle cells (*Figure 1—figure supplement 1*). Control tissue was obtained at the same level of brainstem but outside the region of interest. Each population of cells was pooled from three to eight adult mice per region for subsequent quantitative RT-PCR for each of the fourteen murine P2 receptor subtypes. The type and relative abundance of P2 receptor transcript expressed by each cell type was similar to the cortex (*Vanlandewijck et al., 2018*) and relatively consistent between brainstem regions. For example, endothelial cells from each region express ionotropic *P2rx4* and *P2rx7* and metabotropic *P2ry1,2,6,12-14* receptor transcript, albeit at varying levels relative to control (*Figure 1A*). Also, *P2rx1,4* and *P2ry2,6,14* receptor transcript were detected in smooth muscle cells from each region and most were under-expressed relative to control (*Figure 1B*). The P2 transcript profile of vascular cells in the RTN was fairly similar to the ROb (*Figure 1A–B*) but differed from the cNTS in several ways including lower expression of *P2ry6* in endothelial cells (*Figure 1A*) and higher expression of *P2rx1*, *P2rx4*, and *P2ry14* in smooth muscle cells (*Figure 1B*). However, only *P2ry2* in the RTN showed a distinct expression profile that was consistent across cell types with a role in vasoconstriction; *P2ry2* transcript was detected at lower ($-9.2 \pm 0.3$ $\log_2$ fold change, p=0.0013) and higher ($6.8 \pm 1.5$ $\log_2$ fold change, p=0.0462) levels compared to control in RTN endothelial and vascular smooth muscle cells, respectively (*Figure 1A–B*). This is interesting because in other brain regions activation of endothelial $P2Y_2$ receptors favors nitric-oxide-induced vasodilation (*Marrelli, 2001*), whereas activation of this same receptor in vascular smooth muscle cells mediates vasoconstriction (*Brayden et al., 2013*; *Lewis et al., 2000*). These results identify $P2Y_2$ receptors in vascular smooth muscle cells as potential substrates for $CO_2/H^+$-induced ATP-dependent vasoconstriction in the RTN.

### $CO_2/H^+$ differentially effects vascular tone in the RTN by a $P2Y_2$-dependent mechanism

To further test this possibility, we used the brain slice preparation, optimized for detecting increases or decreases in vascular tone, to characterize $CO_2/H^+$ vascular responses in each chemoreceptor region of interest as well in a non-respiratory region of the somatosensory cortex. For these experiments, we target arterioles based on previously described criteria (*Filosa et al., 2004*; *Mishra et al., 2014*). We verified that exposure to $CO_2/H^+$ (15% $CO_2$; pH 6.9) under control conditions decreases

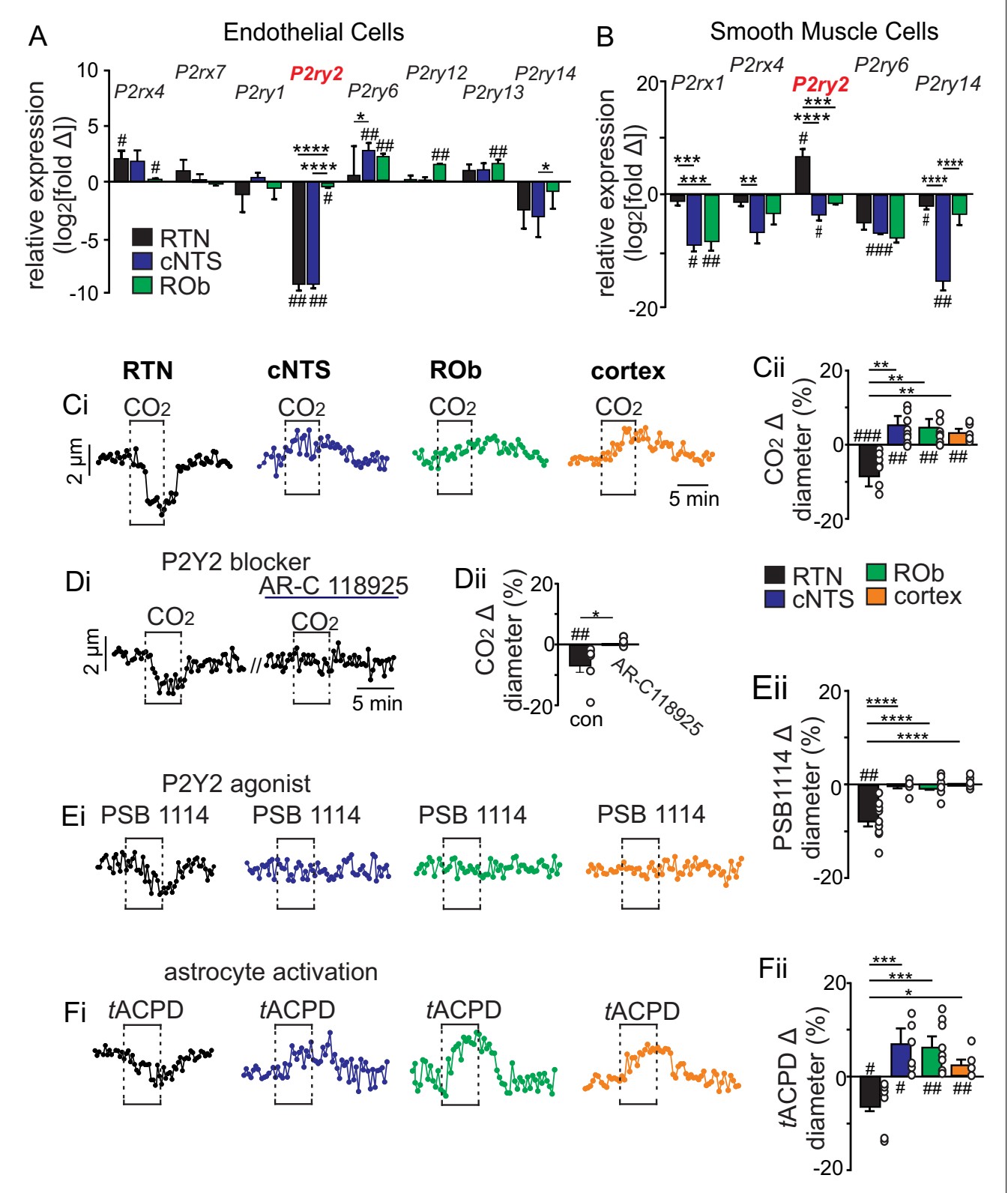

**Figure 1.** $CO_2/H^+$ differentially regulates arteriole tone in the RTN by a $P2Y_2$ receptor dependent mechanism. (**A-B**) Tissue sections containing the cNTS, ROb, and RTN were prepared from an endothelial cell ($Tek^{Cre}$::TdTomato) and smooth muscle cell reporter mice ($Myh11^{Cre/eGFP}$). Individual cells were dissociated and sorted to isolate enriched cell populations from each region (100–300 cells/region). Control cells for each region were prepared from slices with experimental regions removed. Fold change of each P2 receptor was determined for each region by normalizing to within group

*Figure 1 continued on next page*

Figure 1 continued

*Gapdh* expression as well as to control group receptor expression and plotted as log$_2$[fold change]; 0 on the y-axis indicates control group expression for each receptor and negative values reflect less than the control group and positive values reflect greater than control group. Of all P2 receptors detected in each region, only *P2ry2* showed an expression pattern consistent with a role in vasoconstriction in the RTN; low expression in endothelial cells (A) and above baseline expression in smooth muscle cells (B) from this region. (C-F) Diameter of arterioles in the RTN, cNTS, ROb and somatosenstory cortex was monitored in brain slices from adult mice over time by fluorescent video microscopy. (C-F) Diameter traces of individual arterioles in each region and corresponding summary bar graphs show that exposure to 15% $CO_2$ (Ci-ii), activation of P2Y$_2$ receptors by bath application of PSB1114 (100 µM) (Ei-ii), or activation of astrocytes by bath application of t-ACPD (50 µM) (Fi-ii) caused vasoconstriction in the RTN and dilation in all other regions of interest. (Di-Dii) The $CO_2/H^+$ response of RTN vessels was blocked by a selective P2Y$_2$ receptor antagonist (AR-C118925; 10 µM) #, difference from baseline (one sample t-test). *, differences in each condition (one-way ANOVA with Tukey's multiple comparison test). One symbol = p < 0.05, two symbols = p < 0.01, three symbols = p < 0.001, four symbols = p < 0.0001.

The online version of this article includes the following source data and figure supplement(s) for figure 1:

**Source data 1.** Raw Ct values of P2 purinergic transcripts and control marker expression for endothelial and smooth muscle cells.
**Figure supplement 1.** Gating Strategy for FACS sorting of smooth muscle cell and endothelial cell populations from a single-cell suspension.
**Figure supplement 2.** $CO_2/H^+$-induced constriction of RTN arterioles in vitro is not dependent on neural activity, prostaglandin EP$_3$ receptors or adenosine signaling.

the diameter of RTN arterioles by $-11.2 \pm 4.5\%$ (N = 9 vessels), whereas this same stimulus consistently dilates arterioles in the cNTS ($\Delta$ +6.0 $\pm$ 2.2%; N = 9 vessels), ROb ($\Delta$ +6.5 $\pm$ 2.6%; N = 11 vessels) and somatosensory cortex ($\Delta$ +3.7 $\pm$ 0.9%; N = 9 vessels) ($F_{3,34}$=13.49; p<0.0001) (*Figure 1Ci–Cii*). Consistent with a requisite role of P2Y$_2$ receptors in this response, we found that $CO_2/H^+$-induced constriction of RTN arterioles was eliminated by blockade of P2Y$_2$ receptors with AR-C118925 (10 µM) ($\Delta$ +0.2 $\pm$ 0.3%; N = 6 vessels) ($T_5$ = 1.223; p>0.05) (*Figure 1Di–Dii*) and mimicked by application of a selective P2Y$_2$ receptor agonist (PSB 1114; 200 nM) ($\Delta$ $-2.3 \pm 0.7\%$; N = 11 vessels) ($T_{10}$ = 3.460; p<0.010) (*Figure 1Ei–Eii*). As a negative control, we confirmed that PSB 1114 (200 nM) minimally affects vascular tone in the cNTS ($\Delta$ $-0.05 \pm 0.09\%$; N = 6 vessels; $T_5$ = 0.3898; p>0.05) and ROb ($\Delta$ $-0.07 \pm 0.19\%$; N = 7 vessels; $T_6$ = 0.6076; p>0.05) (*Figure 1Ei–Eii*), where P2Y$_2$ receptors were detected at low levels in both endothelial and smooth muscle cells (*Figure 1A–B*). We also confirmed that the response of RTN arterioles to $CO_2/H^+$ was retained when neural action potentials were blocked by bath application of TTX (0.5 µM) (*Figure 1—figure supplement 2*), indicating that $CO_2/H^+$ constriction of RTN arterioles differs from canonical neurovascular coupling both in terms of polarity and dependence (or lack thereof) on neuronal activity (*Ladecola, 2017*). Also, despite evidence that extracellular ATP can be rapidly metabolized to adenosine (a potent vasodilator including in the RTN [*Hawkins et al., 2017*]) the response of RTN arterioles to $CO_2/H^+$ was unaffected by 8-phenyltheophylline (8 PT; 10 µM) to block adenosine A1 receptors or sodium metatungstate (POM 1; 100 µM) to inhibit ectonucleotidase activity (*Figure 1—figure supplement 2*). Furthermore, exposure to $CO_2/H^+$ has been shown to trigger prostaglandin E$_2$ (PGE$_2$) release that may contribute to RTN chemoreception by a mechanism involving EP$_3$ receptors (*Forsberg et al., 2016*); therefore, we also tested for a role of PGE$_2$/EP$_3$ signaling in $CO_2/H^+$-dependent regulation of RTN arteriole tone. We found bath application of PGE$_2$ (1 µM) decreased RTN arteriole tone by $-2.7 \pm 0.4\%$ ($T_9$ = 2.787, p=0.0212, data not shown). However, $CO_2/H^+$ constriction of RTN vessels was retained in the presence of L-798,106 (0.5 µM) to block EP$_3$ receptors ($\Delta$ $-9.6 \pm 4.0\%$, $T_7$ = 2.675, p=0.0318) (*Figure 1—figure supplement 2*), suggesting PGE$_2$/EP$_3$ signaling is dispensable for $CO_2/H^+$ regulation of arteriole tone in the RTN.

Astrocytes are thought to contribute to $CO_2/H^+$ induced vasodilation in the cortex (*Howarth et al., 2017*) and vasoconstriction in the RTN (*Hawkins et al., 2017*). Consistent with this, we found that exposure to t-ACPD (50 µM), an mGluR agonist widely used to elicit Ca$^{2+}$ elevations in astrocytes (*Howarth et al., 2017*), caused constriction of RTN arterioles ($\Delta$ $-5.9 \pm 1.7\%$; N = 8 vessels) and dilation of arterioles in the cNTS ($\Delta$ +7.6 $\pm$ 2.5%; N = 8 vessels), ROb ($\Delta$ +6.8 $\pm$ 1.9%; N = 8 vessels) and somatosensory cortex ($\Delta$ +2.9 $\pm$ 0.8%; N = 5 vessels) ($F_{3,25}$=10.18; p<0.0001) (*Figure 1Fi–Fii*). These results show that $CO_2/H^+$-dependent regulation of vascular tone and roles of astrocytes in this process are fundamentally different in the RTN compared to other brain regions, and point to differential purinergic signaling mechanisms downstream of astrocyte activation.

To determine whether P2Y$_2$ receptors in the RTN regulate $CO_2/H^+$ vascular reactivity in vivo, we measured the diameter of pial vessels on the ventral medullary surface (VMS) in the region of the

RTN during exposure to high $CO_2$ under control conditions (saline) and when P2Y$_2$ receptor are blocked with AR-C118925. Consistent with our slice data, we found under saline control conditions that increasing end-expiratory $CO_2$ to 9–10%, which corresponds with an arterial pH of 7.2 pH units (*Guyenet et al., 2005*), constricted VMS vessels by $-7.8 \pm 0.7\%$ (p=0.01, N = 7 animals) (*Figure 2A*). However, when P2Y$_2$-receptors are blocked by application of AR-C118925 (10 µM) to the VMS, increasing inspired $CO_2$ resulted in a vasodilation of $5.2 \pm 0.9\%$ (*Figure 2A*) (p=0.05; N = 7 animals). Thus, in the absence of functional P2Y$_2$ receptors, RTN vessels respond to $CO_2/H^+$ in a manner similar to other brain regions. Also consistent with our slice data, we found that activation of P2Y$_2$ receptors by application of PSB1114 (100 µM) to the VMS constricted vessels in the region of the RTN by $-6.3 \pm 0.9\%$ (p=0.05, N = 7 animals) (*Figure 2B*).

To correlate P2Y$_2$-dependent vasoconstriction in the RTN region with respiratory behavior, we simultaneously measured systemic blood pressure and external intercostal electromyogram (Int$_{EMG}$) activity (as a measure of respiratory activity) in urethane-anesthetized mice during exposure to low and high $CO_2$ following VMS application of saline or AR-C118925. We found that VMS application of AR-C118925 (1 mM) minimally affected the $CO_2/H^+$ apneic threshold ($3.3 \pm 0.4\%$ vs. saline: $3.1 \pm 0.5\%$) (p>0.05; two-way RM ANOVA; N = 7). However, at high levels of $CO_2$ (9–10% etCO$_2$) AR-C118925 decreased intercostal EMG amplitude by $23 \pm 5\%$ ($F_{2,74} = 69.83$; p=0.021; N = 7 animals) (*Figure 2C,E*), but with no change in frequency ($F_{2,74} = 1.09$; p>0.05; N = 7 animals) (*Figure 2C,F*). Conversely, VMS application of PSB1114 (100 µM) while holding etCO$_2$ constant at 5% increased intercostal EMG amplitude by $20 \pm 3\%$ ($F_{2,74} = 96.14$; p=0.02; N = 7 animals) (*Figure 2D,H*), again with no change in frequency ($F_{2,74} = 0.79$; p>0.05; N = 7 animals) (*Figure 2D,I*). These treatments had negligible effects on systemic mean arterial pressure (MAP) (AR-C118925: $113 \pm 5$; PSB1114: $112 \pm 5$; saline: $114 \pm 4$ mmHg; $F_{2,74} = 1.29$; p>0.05; N = 7 animals) (*Figure 2G,J*). These results indicate that P2Y$_2$ dependent vasoconstriction in the RTN contributes to the drive to breathe.

If the mechanism underlying vascular control of RTN chemoreception involves maintenance of tissue $CO_2/H^+$ levels, then by this same logic, vasodilation in the cNTS and ROb should buffer tissue $CO_2/H^+$ and limit contributions of these regions to respiratory output. Such a braking mechanism may be important for stabilizing breathing since over-activation of the $CO_2/H^+$ chemoreflex is thought to cause unstable periodic breathing (*Cherniack and Longobardo, 2006*). To test this, we disrupted $CO_2/H^+$ dilation by injecting the vasoconstrictor U46619 (a thromboxane A2 receptor agonist) into the cNTS and ROb while measuring cardiorespiratory activity in anesthetized mice breathing a level of $CO_2$ required to maintain respiratory activity (2–3% $CO_2$). We found that injections of U46619 (1 µM) into the cNTS alone caused an increase in breathing frequency but together with the ROb resulted in unstable breathing as evidenced by frequent bouts of hyperventilation followed by apnea (*Figure 3A*) ($1.2 \pm 0.6$, vs. saline: $0.4 \pm 0.2$ apneas/min; p<0.001) and increased breath to breath variability (*Figure 3C–E*). Consistent with increased chemoreceptor drive, we found that application of U46619 into these regions lowered the $CO_2$ apneic threshold from $3.2 \pm 0.3\%$ to $2.1 \pm 0.1\%$ (p<0.05; two-way RM ANOVA; N = 7) (*Figure 3F*). Also, consistent with previous work (*Yalcin and Savci, 2004*), we found that injection of U46619 (1 µM) into the cNTS also increased systemic blood pressure ($127 \pm 11$, vs. saline: $98 \pm 4$ mmHg), ($F_{3,65} = 77.62$; p<0.01, data not shown), whereas, application of this drug into the ROb alone had negligible effects on cardiorespiratory output. Together, these results show that $CO_2/H^+$ induced vasoconstriction in the RTN serves to enhance chemoreceptor drive, while simultaneous $CO_2/H^+$-dependent dilation in the cNTS and possibly the ROb limits chemoreceptor activity to stabilize $CO_2/H^+$ stimulated respiratory drive.

## Smooth muscle specific P2Y$_2$ cKO mice have a blunted chemoreflex that can be rescued by targeted re-expression of P2Y$_2$ in RTN smooth muscle cells

To definitively test contributions of P2Y$_2$ receptors in vascular smooth muscle cells to respiratory activity, we created a smooth muscle cell-specific P2Y$_2$ knockout mouse (P2Y$_2$ cKO maintained on a C57BL6/J background) by crossing *Tagln*$^{Cre}$ (JAX #: 017491) with *P2ry2*$^{f/f}$ mice provided by Dr. Cheike Seye (Indiana Univ.). We confirmed that Cre recombinase expression was restricted to vascular smooth muscle cells (*Figure 4—source data 1*) and that *P2ry2* transcript was not detectable in RTN smooth muscle cells from P2Y$_2$ cKO mice (*Figure 4A*). Each genotype (*Tagln*$^{Cre}$ only, *P2ry2*$^{f/f}$ only and P2Y$_2$ cKO mice) was obtained at the expected ratios and gross motor activity, metabolic activity, blood gases, heart rate and blood pressure were all similar between genotypes and sexes

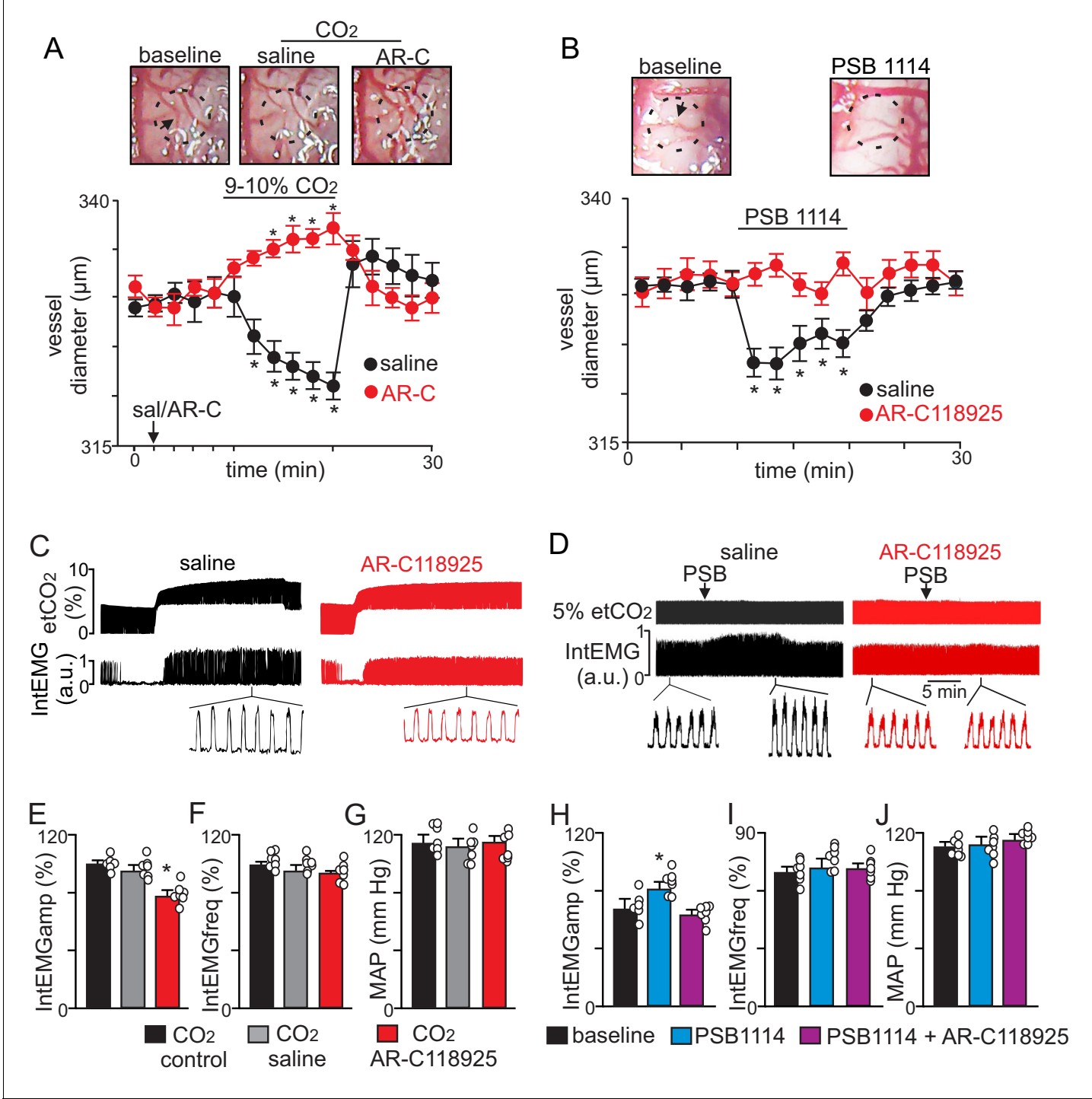

**Figure 2.** $CO_2$ constricts pial vessels in the RTN region by a P2Y$_2$ receptor-dependent mechanism to increase respiratory behavior in anesthetized mice. (**A**) Images of RTN pial vessels and corresponding traces of vessel diameter (N = 6 mice/condition) show that exposure to 9–10% $CO_2$ decreased vessel diameter under control conditions (saline) but not when P2Y$_2$ receptors were blocked with AR-C118925 (1 mM). (**B**) Images of RTN pial vessels and corresponding traces vessel diameter show that application of a P2Y$_2$ receptor agonist (PSB1114; 100 µM) caused a reversible constriction. (**C-D**) Traces of external intercostal EMG (Int$_{EMG}$) and end expiratory $CO_2$ (et$CO_2$) show that blocking $CO_2$/H$^+$-induced vasoconstriction by ventral surface application of AR-C118925 minimally affected respiratory activity at low et$CO_2$ levels but blunted the ventilatory response to 9–10% $CO_2$ (**C**). Conversely, at a constant et$CO_2$ of 5% the application of PSB1114 to mimic $CO_2$/H$^+$ constriction increased respiratory output (**D**). (**E-J**) Summary data show (N = 6 mice/condition) effects RTN application of saline, AR-C118925 or PSB1114 on intercostal EMG amplitude (**E, H**), frequency (**F, I**) and mean arterial pressure (MAP; **G, J**). *, Different (RM-ANOVA followed by Bonferroni multiple-comparison test; *, $p < 0.05$). scale bar = 200 µm.

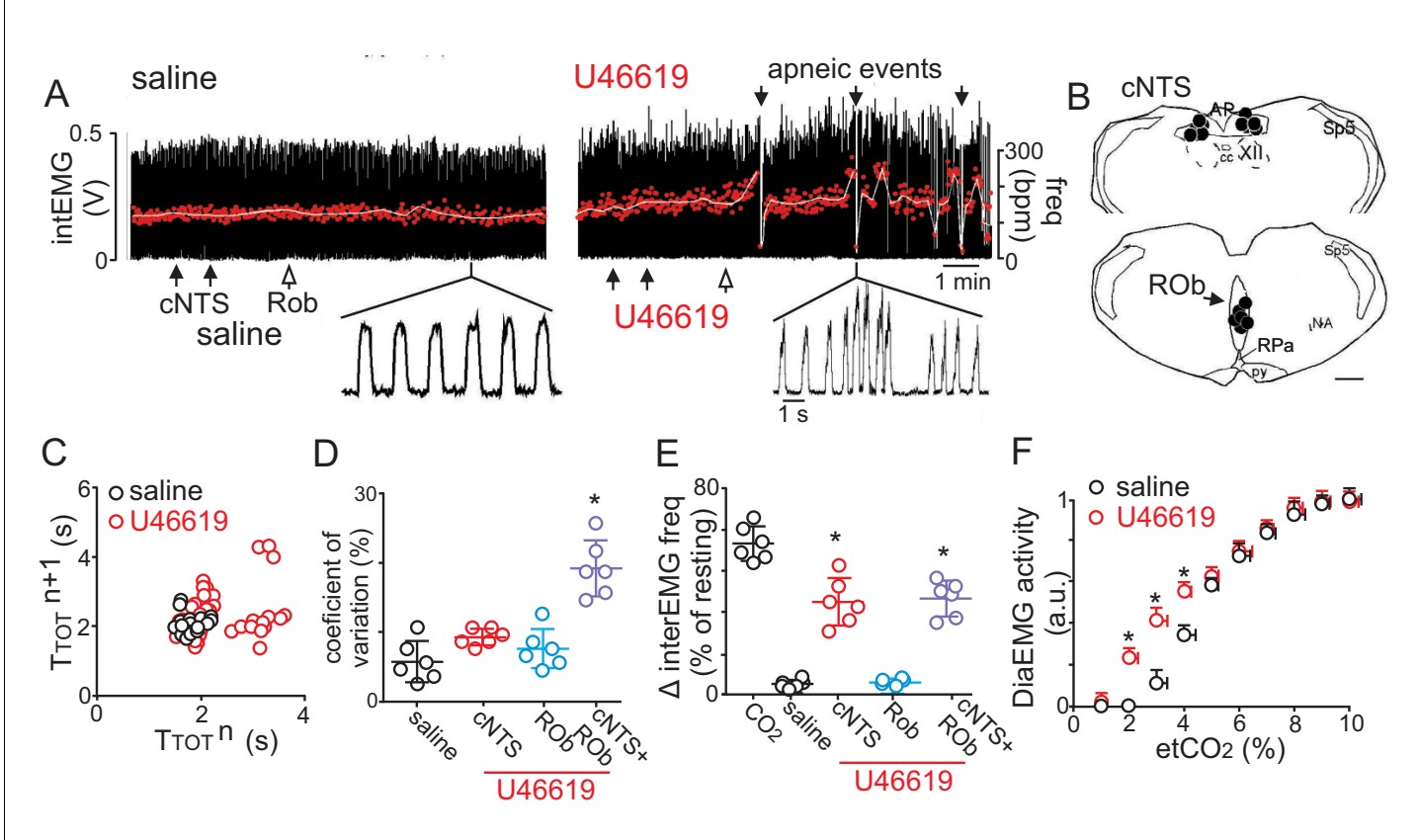

**Figure 3.** Disruption of $CO_2/H^+$ dilation in the cNTS and ROb causes unstable breathing and apnea. (**A**) Trace of external intercostal muscle EMG ($Int_{EMG}$) activity shows respiratory activity of an anesthetized wild type mouse breathing 2.5% $CO_2$ following injections of saline or U46619 (1 µM; 30 nL/ region) into the cNTS and ROb. (**B**) Location of injections in the cNTS and ROb. (**C**) Representative Poincaré plot (50 breaths) shows breath-to-breath ($T_{TOT}$) interval variability following injections saline (black) or U46619 (red) conditions. (**D-E**) Summary data (N = 6 animals/group) shows effects of U46619 injections into the cNTS and ROb alone and in combination on the coefficient of variation of $Int_{EMG}$ frequency (**C**) and $Int_{EMG}$ frequency (**E**). (**F**) Summary data show that injections of U46619 injections into the cNTS and ROb lowered the $CO_2$ apneic threshold from 3.2 ± 0.3% to 2.1 ± 0.1% (N = 7 mice). *, Difference in $Int_{EMG}$ activity under control conditions (saline) vs. during U46119 into the NTS and/or ROb (RM-ANOVA followed by Bonferroni multiple-comparison test, p<0.05). scale bar = 200 µm.

(**Table 1**, **Figure 4—figure supplement 1**). Therefore, results from male and female mice of each genotype were pooled. To determine whether P2Y$_2$ cKO mice exhibit respiratory problems, we used whole-body plethysmography to measure baseline breathing and the ventilatory response to $CO_2$ in conscious unrestrained adult mice (mixed sex). Consistent with the possibly that smooth muscle P2Y$_2$ receptor-mediated vasoconstriction in the RTN augments the ventilatory response to $CO_2$, we found that P2Y$_2$ cKO mice exhibit normal respiratory activity under room air conditions (**Table 1**) but pronounced hypoventilation during exposure to graded increases in $CO_2$ (balance $O_2$ to minimize input from peripheral chemoreceptors). For example, minute ventilation – the product of frequency and tidal volume – of P2Y$_2$ cKO mice at 5% and 7% $CO_2$ was 28% and 35% smaller than control counterparts (**Figure 4C–F**, **Table 2**). This chemoreceptor deficit involves a diminished capacity to increase both respiratory frequency (**Figure 4D**) and tidal volume (**Figure 4E**) during exposure to $CO_2$ (**Table 2**). Similar results were obtained in P2Y$_2$ cKO mice generated using a different smooth muscle cre line (*Myh11*$^{Cre/eGFP}$) (**Figure 4—figure supplement 2**). Also, smooth muscle P2Y$_2$ cKO mice showed a normal ventilatory response to hypoxia (10% $O_2$; balance $N_2$, $T_7$ = 0.1089, p>0.05) (**Figure 4—figure supplement 3**). We confirmed in vitro that RTN arterioles in slices from P2Y$_2$ cKO do not respond to PBS 1114 (0.4 ± 0.1%, $T_7$ = 1661, p>0.05, N = 8 vessels). Interestingly, we also found that RTN arterioles in slices from P2Y$_2$ cKO mice fail to constrict in response to $CO_2/H^+$ or tACPD, but rather respond in a manner similar to arterioles in other brain regions. For example, both $CO_2/H^+$ and tACPD dilated RTN ($CO_2/H^+$: 4.7 ± 0.2%, $T_9$ = 3.312, p=0.009, N = 10 vessels;

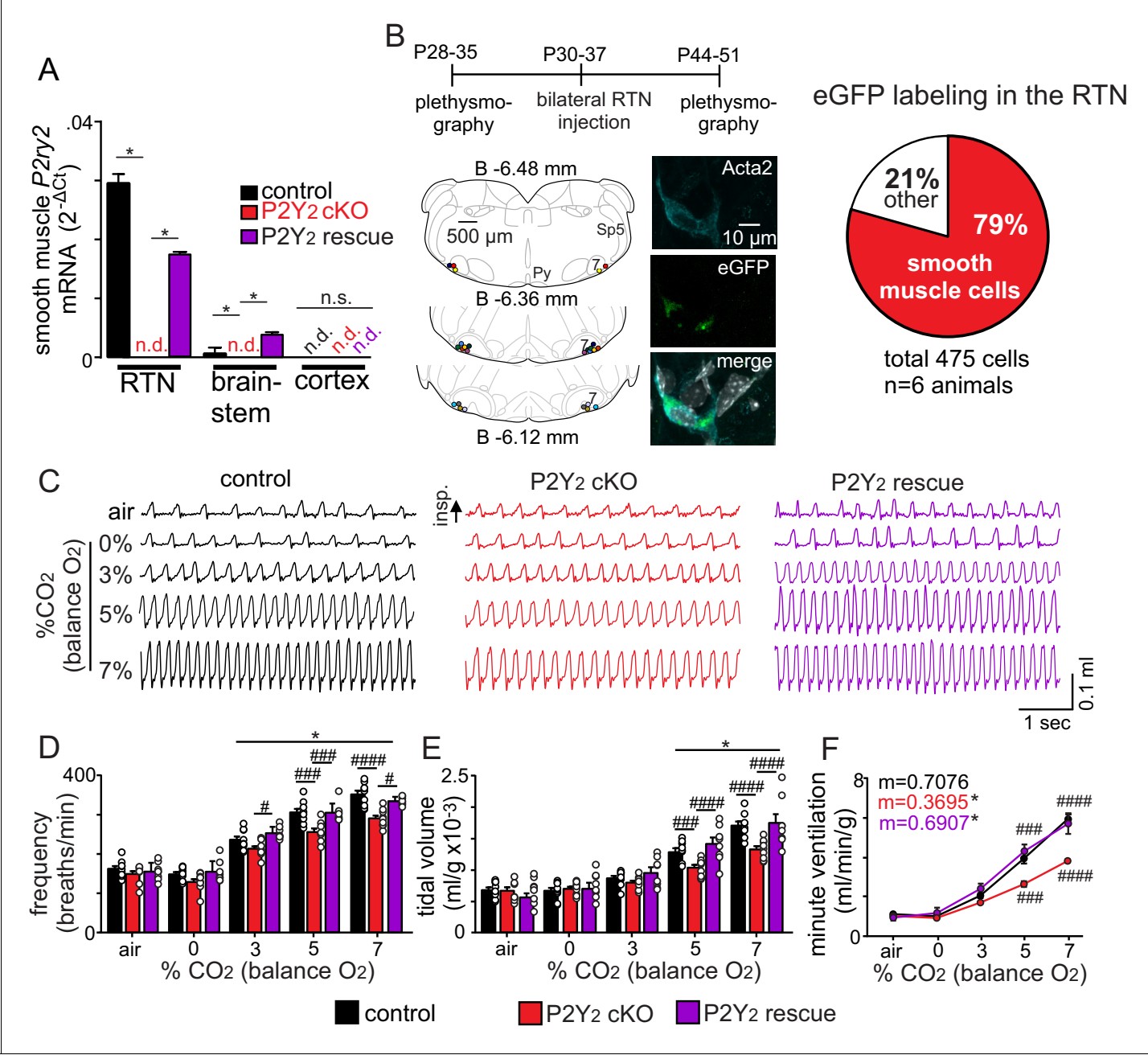

**Figure 4.** Smooth muscle $P2Y_2$ cKO mice show a blunted $CO_2$ chemoreflex that can be rescued by re-expression of $P2Y_2$ only in RTN smooth muscle cells. (**A**) $P2Y_2$ transcript was detected in RTN, brainstem and cortical smooth muscle cells isolated from control mice (*Tagln*[Cre]::TdTomato), $P2Y_2$ cKO mice (*Tagln*[Cre]::*P2ry2*[f/f]::TdTomato), and $P2Y_2$ rescue mice ($P2Y_2$ cKO animals that received bilateral RTN injections of AAV2-*Myh11*p-eGFP-2A-m*P2ry2*). *P2ry2* transcript was not detected (n.d.) in smooth muscle cells from $P2Y_2$ cKO mice (N = 3 runs/9 animals). *P2ry2* was also not detected in cortical smooth muscle cells from either genotype. Conversely, RTN (p=0.0073) and brainstem (p=0.0073) but not cortical (p>0.05) smooth muscle cells from $P2Y_2$ rescue mice show increased *P2ry2* transcript compared to $P2Y_2$ cKO but not to the same level as cells from control mice (p=0.0219) (ANOVA on ranks followed by Dunn multiple comparison test). (**B**) Left, computer-assisted plots show the center of all bilateral AAV2-*Myh11*p-eGFP-2A-m*P2ry2* injections; each matching color pair of dots corresponds to one animal (N = 13 animals). Approximate millimeters behind bregma (***Paxinos and Franklin, 2013***) is indicated by numbers next to each section. Right, 2 weeks after injections we confirmed that ~80% of RTN smooth muscle α2 actin (Acta2)-immunoreactive cells were also GFP[+](inset). (**C-F**) Representative traces of respiratory activity (**C**) and summary data show that smooth muscle-specific $P2Y_2$ KO mice (*Tagln*[Cre]::*P2ry2*[f/f]) breathe normally under room air conditions but fail to increase respiratory frequency (**C**) or tidal volume (**D**) during exposure to $CO_2$, thus resulting in diminished minute ventilation at 5–7% $CO_2$ (**E**). Re-expression of $P2Y_2$ in only RTN smooth muscle cells rescued of the ventilatory response to $CO_2$. Note that *Tagln*[Cre] only and *P2ry2*[f/f] only control mice show similar baseline breathing and responses to

*Figure 4 continued on next page*

*Figure 4 continued*

$CO_2$ and so were pooled. *, Different from 0% $CO_2$ in condition as assessed by Tukey's post-hoc multiple comparison test. ####, Different between genotypes (two-way ANOVA with Tukey's multiple comparison test, p<0.0001).

The online version of this article includes the following source data and figure supplement(s) for figure 4:

**Source data 1.** Raw Ct values of P2Y$_2$ cKO and P2Y$_2$ rescue mice.
**Source data 2.** Metabolic testing in light and dark cycles of control and P2Y$_2$ cKO mice.
**Figure supplement 1.** Cardiovascular, metabolic and blood gas parameters in control and smooth muscle P2Y$_2$ cKO mice.
**Figure supplement 2.** Respiratory activity of *Myh11*$^{Cre/eGFP}$::*P2ry2*$^{f/f}$ and control viral injected P2Y$_2$ cKO mice.
**Figure supplement 3.** P2Y$_2$ cKO mice show a normal ventilatory response to acute hypoxia.
**Figure supplement 4.** Functional characterization of RTN and cortical arterioles in slices from P2Y$_2$ cKO mice.
**Figure supplement 5.** Cell specificity of AAV2-*Myh11*p-eGFP-2A-mP2ry2 transduction.

tACPD: 3.2 ± 0.7%, $T_7$ = 2.864, p=0.035, N = 8 vessels) and cortical ($CO_2/H^+$: 7.3 ± 1.5%, $T_6$ = 4.307, p=0.0051, N = 7 vessels; tACPD: 6.7 ± 0.9%, $T_6$ = 2.739, p=0.0338, N = 7 vessels) arterioles in slices from P2Y$_2$ cKO mice (*Figure 4—figure supplement 4*). This suggests $CO_2/H^+$ induced vasodilation is a general phenomenon mediated by undetermined mechanisms, but in the RTN is response is countered by smooth muscle P2Y$_2$ receptor-mediated constriction.

We next tested whether targeted re-expression of P2Y$_2$ in RTN smooth muscle cells would rescue the blunted chemoreflex in P2Y$_2$ cKO mice. An adeno-associated virus (AAV2-*Myh11*p-eGFP-2A-m*P2ry2*, Vector Biolabs) was injected bilaterally into the RTN of adult P2Y$_2$ cKO (*Tagln*$^{Cre}$::*P2ry2*$^{f/f}$) to drive P2Y$_2$ expression in smooth muscle cells. Two weeks after virus injection into the RTN region, we confirmed that ~80% of smooth muscle α2 actin (Acta2)-immunoreactive cells in the RTN are also GFP$^+$ (*Figure 4—figure supplement 5*). Some background eGFP labeling was observed, but not associated with DAPI labeling (*Figure 4—figure supplement 5*). We also confirmed RTN smooth muscle cells from P2Y$_2$ rescue mice show increased *P2ry2* transcript levels compared to P2Y$_2$ cKO (p=0.0073) but not to the same level as cells from control mice ($H_3$ = 7.2, p=0.0219) (*Figure 4A*). Importantly, re-expression of P2Y$_2$ receptors in RTN smooth muscle cells resulted in a full rescue of the minute ventilatory response to $CO_2$ (*Figure 4C–F*, *Table 2*) ($F_{1,7}$=17.75, p=0.004, N = 13 animals). Conversely, bilateral RTN injections of control virus (AAV2-*Myh11*p-eGFP, Vector Biolabs) minimally affected respiratory parameters of interest in P2Y$_2$ cKO mice (*Figure 4—figure supplement 2*). Furthermore, re-expression of P2Y$_2$ receptors in RTN smooth muscle cells also rescues $CO_2/H^+$ vascular reactivity. Specifically, we found that, in slices from P2Y$_2$ cKO mice, RTN arterioles

**Table 1.** Respiratory parameters in control (*Tagln*$^{Cre}$ only and *P2ry2*$^{f/f}$ only), P2Y$_2$ cKO (*Tagln*$^{Cre}$::*P2ry2*$^{f/f}$) and P2Y$_2$ rescue animals 2 weeks after bilateral RTN injections of AAV2-*Myh11*p-eGFP-2A-m*P2ry2*.
No significant differences were observed (p>0.05).

| | *Tagln*$^{Cre}$ only, *P2ry2*$^{f/f}$ only | P2Y$_2$ cKO |
|---|---|---|
| Frequency (breaths/min) | 163.4 ± 7.1 | 149.8 ± 7.2 |
| Tidal volume (mL/g x 10$^{-3}$) | 6.8 ± 0.4 | 6.7 ± 0.5 |
| Minute ventilation (mL/min/g) | 1.1 ± 0.1 | 1.0 ± 0.1 |
| Systolic blood pressure (mmHg) | 124.2 ± 0.9 | 123.0 ± 1.2 |
| Diastolic blood pressure (mmHg) | 85.5 ± 0.8 | 86.7 ± 1.4 |
| Heart rate (beats/min) | 473.8 ± 3.0 | 465.0 ± 5.0 |
| VO$_2$ (mL/kg/hr) | 4886.5 ± 127.1 | 5067.0 ± 230.4 |
| VCO$_2$ (mL/kg/hr) | 4392.8 ± 101.9 | 4653.8 ± 204.5 |
| Respiratory exchange ratio (RER) | 0.89 ± 0.01 | 0.91 ± 0.01 |
| pH | 7.410 ± 0.005 | 7.406 ± 0.008 |
| pCO$_2$ (mmHg) | 40.1 ± 0.4 | 40.7 ± 0.6 |
| pO$_2$ (mmHg) | 88.3 ± 0.9 | 88.0 ± 0.8 |

**Table 2.** Cardiorespiratory and metabolic parameters under room air conditions in control and P2Y$_2$ cKO mice.

| A | *Tagln*$^{Cre}$ or *P2ry2*$^{f/f}$ only | | |
|---|---|---|---|
|  | Frequency (breaths/minute) | Tidal volume (mL/g x 10$^{-3}$) | Minute ventilation (mL/min/g) |
| Room air | 163.4 ± 7.1 | 6.8 ± 0.4 | 1.1 ± 0.1 |
| 0% CO$_2$/100% O$_2$ | 149.1 ± 5.7 | 6.7 ± 0.5 | 1.0 ± 0.1 |
| 3% CO$_2$/97% O$_2$ | 236.6 ± 8.5 **** | 8.7 ± 0.4 **** | 2.1 ± 0.1 **** |
| 5% CO$_2$/95% O$_2$ | 306.0 ± 9.4 **** | 12.9 ± 0.6 **** | 4.0 ± 0.3 **** |
| 7% CO$_2$/93% O$_2$ | 351.6 ± 9.8 **** | 17.1 ± 1.4 **** | 6.0 ± 0.3 **** |
| B | P2Y$_2$ cKO | | |
|  | Frequency (breaths/minute) | Tidal volume (mL/g x 10$^{-3}$) | Minute ventilation (mL/min/g) |
| Room air | 149.8 ± 7.2 | 6.7 ± 0.5 | 1.0 ± 0.1 |
| 0% CO$_2$/100% O$_2$ | 130.1 ± 7.4 | 7.1 ± 0.3 | 0.9 ± 0.1 |
| 3% CO$_2$/97% O$_2$ | 213.7 ± 6.5 **** | 8.0 ± 0.3 | 1.7 ± 0.1 ** |
| 5% CO$_2$/95% O$_2$ | 256.2 ± 9.6 ### **** | 10.4 ± 0.4 #### **** | 2.7 ± 0.2 #### **** |
| 7% CO$_2$/93% O$_2$ | 291.3 ± 7.5 #### **** | 13.3 ± 0.5 #### **** | 3.9 ± 0.1 #### **** |
| C | P2Y$_2$ cKO rescue | | |
|  | Frequency (breaths/minute) | Tidal volume (mL/g x 10$^{-3}$) | Minute ventilation (mL/min/g) |
| Room air | 156.3 ± 9.0 | 5.6 ± 0.1 | 0.9 ± 0.2 |
| 0% CO$_2$/100% O$_2$ | 158.3 ± 12.8 | 6.8 ± 0.1 | 1.1 ± 0.3 |
| 3% CO$_2$/97% O$_2$ | 254.0 ± 6.2 ## **** | 9.5 ± 0.1 *** | 2.4 ± 0.3 **** |
| 5% CO$_2$/95% O$_2$ | 305.1 ± 9.8 #### **** | 14.0 ± 0.1 ## **** | 4.3 ± 0.4 #### **** |
| 7% CO$_2$/93% O$_2$ | 335.6 ± 4.1 ### **** | 17.0 ± 0.2 ### **** | 5.7 ± 0.6 #### **** |

*,difference from 0% CO$_2$; #, differences between genotypes under an experimental condition (two-way ANOVA with Tukey's multiple comparison test). Two symbols = p < 0.01, three symbols = p < 0.001, four symbols = p < 0.0001.

transfected with AAV showed a robust constriction in response to CO$_2$ ($-19.9 \pm 6.9\%$, p=0.0421), tACPD ($-6.2 \pm 1.9\%$, p=0.0092) and PSB1114 ($-9.3 \pm 4.9\%$, p=0.0154) (*Figure 4—figure supplement 4*). These results identify the first vascular element of respiratory control by showing that P2Y$_2$ receptors in RTN vascular smooth muscle cells are required for the normal ventilatory response to CO$_2$.

## Discussion

These results identify P2Y$_2$ receptors in RTN smooth muscle cells as a novel vascular element of respiratory chemoreception. We show that CO$_2$/H$^+$ constriction by activation of smooth muscle P2Y$_2$ receptors is unique to the RTN and is required for the normal ventilatory response to CO$_2$, whereas simultaneous CO$_2$/H$^+$ vasodilation in other chemoreceptor regions like the cNTS and ROb may serve to stabilizes breathing. Considering these regions sense changes in CO$_2$/H$^+$ to regulate breathing, and since vasoconstriction and dilation are expected to increase and decrease tissue CO$_2$/H$^+$ levels respectively, we speculate that differential CO$_2$/H$^+$ vascular reactivity across multiple chemoreceptor regions serves to fine tune respiratory drive and stability during exposure to high CO$_2$. Understanding this novel mechanism may lead to new treatment options for breathing problems, particularly those associated with cardiovascular disease.

### Experimental limitations

There are several limitations of this work that should be recognized. First, our in vitro experiments utilized the slice preparation because it allows for easy visualization of vessels in each region of interest and pharmacological manipulation of candidate mechanisms independent of potential

confounding effects of blood pressure on myogenic tone. However, because blood vessels in brain slice have limited myogenic tone these experiments were performed in the presence of U-46619 to pre-constrict vessels ~ 30% (*Blanco et al., 2008*). We also used a large stimulus (15% $CO_2$) to characterize $CO_2/H^+$ vascular reactivity in vitro. For these reasons, we also confirmed in vivo in the absence of U-46619 that more physiological levels of $CO_2$ (9–10%) constricted arterioles in the RTN by a P2Y$_2$-dependent mechanism. Second, we and others (*Yalcin and Savci, 2004*) showed that administration of U46619 into the NTS increased blood pressure. Although baroreceptor activation has a fairly mild effect on respiratory activity (*McMullan et al., 2009*), it remains possible that the baroreflex contributed to respiratory instability observed during this experimental manipulation. Furthermore, neurons and astrocytes may express thromboxane A2 receptors (*Gao et al., 1997*; *Nakahata et al., 1992*) so it is possible respiratory instability caused by injections of U46619 into the cNTS and ROb may involve direct neural or astrocyte activation. Third, $CO_2/H^+$ typically increases minute ventilation by increasing both rate and depth of breathing (*Souza et al., 2019*; *Takakura et al., 2014*), yet ventral surface application of drugs to manipulate P2Y$_2$ receptors only affected tidal volume (*Figure 2E–J*). Therefore, it is possible other nearby respiratory centers were affected in this experiment. However, this issue is mitigated by more targeted genetic approaches showing that deletion of P2Y$_2$ from smooth muscle cells blunted the rate and depth of respiratory responses to $CO_2$, and re-expression of P2Y$_2$ receptors only in RTN smooth muscle cells fully rescued the $CO_2/H^+$ chemoreflex.

## Mechanisms contributing to vascular control of breathing

Blood flow in the brain is controlled by several mechanisms including neural activity, which leads to vasodilation and increased blood flow by a process termed neurovascular coupling (*Ladecola, 2017*). Autoregulation is a mechanism that maintains cerebral blood flow in response to changes in perfusion pressure (*Paulson et al., 1990*). For example, myogenic tone increases with increased perfusion pressure and relaxes with decreased pressure. $CO_2/H^+$ also functions as a potent vasodilator in most brain regions (*Hoiland et al., 2019*) and is perhaps the most potent regulator of cerebrovascular tone since high $CO_2/H^+$ can impair both autoregulation (*Ogoh and Ainslie, 2009*; *Perry et al., 2014*) and neurovascular coupling (*Maggio et al., 2013*; *Maggio et al., 2014*). Since $CO_2/H^+$ is a waste product of metabolism, $CO_2/H^+$ vascular reactivity provides a means of matching local blood flow with tissue metabolic needs. However, at the level of the RTN this need appears secondary to chemoreceptor regulation of respiratory activity. For example, we (*Hawkins et al., 2017*) and others (*Wenzel et al., 2020*) showed that exposure to high $CO_2/H^+$ constricted RTN arterioles, and disruption of this response blunted the ventilatory response to $CO_2$.

The cellular and transmitter basis for $CO_2/H^+$-induced constriction of RTN arterioles appears to involve $CO_2/H^+$-evoked ATP release from local astrocytes. For example, previous work showed that (i) $CO_2/H^+$ caused discrete release of ATP near the RTN region (*Gourine et al., 2005*); (ii) astrocytes in the RTN show $Ca^{2+}$ responses to mild acidification (*Gourine et al., 2010*); and (iii) activation of astrocytes with tACPD, an mGluR agonist widely used to elicit $Ca^{2+}$ elevations in astrocytes (*Howarth et al., 2017*), constricted RTN arterioles by a purinergic-dependent mechanism (*Hawkins et al., 2017*; *Figure 2B*), whereas application of an ATP receptor blocker to the ventral surface blunted $CO_2/H^+$-induced vasoconstriction and the ventilatory response to $CO_2$ (18). Here, we extend this work by showing that $CO_2/H^+$ vascular responses in the RTN are opposite to other chemoreceptor regions, and we identify P2Y$_2$ receptors in vascular smooth muscle cells as requisite determinants of RTN $CO_2/H^+$ vascular reactivity and chemoreception. For example, P2Y$_2$ is a metabotropic receptor that couples with Gq, Go and G12 proteins (*Erb and Weisman, 2012*; activation of this receptor in smooth muscle is associated with vasoconstriction; *Brayden et al., 2013*; *Lewis et al., 2000*), whereas its activation in endothelial cells favors nitric oxide-mediated vasodilation (*Marrelli, 2001*). Consistent with a role in vasoconstriction, we found that P2Y$_2$ transcript was expressed at higher than control levels in RTN smooth muscle cells and at lower than control levels in RTN endothelial cells (*Figure 1A–B*). Furthermore, pharmacological experiments show in vitro (*Figure 1C–E*) and in anesthetized mice (*Figure 2A–B*) that $CO_2/H^+$-induced constriction of RTN arterioles was eliminated by blocking P2Y$_2$ receptors with AR-C118925 and mimicked by a selective P2Y$_2$ receptor agonist (PSB 1114). We also show in awake mice that genetic deletion of P2Y$_2$ from smooth muscle cells blunted the ventilatory response to $CO_2$, and re-expression of P2Y$_2$ receptors only in RTN smooth muscle cells fully rescued the $CO_2/H^+$ chemoreflex (*Figure 4A–F*). These results

establish $P2Y_2$ receptors in RTN smooth muscle cells as requisite determinants of respiratory chemoreception.

It should be noted that other cell types and signaling pathways may also contribute to RTN vascular reactivity. For example, endothelial cells including those in the RTN, express $H^+$ activated GPR4 receptors that when activated facilitates release of vasoactive effectors including thromboxane to mediate vasoconstriction (*Wenzel et al., 2020*). Consistent with this, disruption of GPR4 signaling blunted $CO_2/H^+$ vascular reactivity in the RTN and the ventilatory response to $CO_2$ (*Wenzel et al., 2020*). Although untested, it is also possible RTN endothelial cells release other vasoactive signals including ATP, and so may contribute to $CO_2/H^+$-evoked purinergic-dependent constriction of arterioles in this region. Prostaglandin $E_2$ ($PGE_2$) is also thought to contribute to $CO_2/H^+$ vascular reactivity. For example, in the cortex $CO_2/H^+$ facilitates $Ca^{2+}$ oscillations in astrocytes (*Gourine et al., 2010*; *Howarth et al., 2017*) leading to enhanced arachidonic acid metabolism and release of $PGE_2$, which triggered arteriole dilation by activation of prostaglandin ($EP_1$) receptors (*Howarth et al., 2017*). In the RTN, $CO_2/H^+$ also causes $PGE_2$ release most likely from astrocytes (*Forsberg et al., 2017*) and contributes to RTN chemoreception by a mechanism involving EP3 receptors (*Forsberg et al., 2016*). However, selective blockade of $EP_3$ receptors minimally affected $CO_2/H^+$ constriction of RTN arterioles (*Figure 1—figure supplement 2*), suggesting $PGE_2/EP_3$ signaling contributes to RTN chemoreception by directly targeting $CO_2/H^+$-sensitive neurons or astrocytes.

In contrast to the RTN, we also found that $CO_2/H^+$ (*Figure 1C*) and astrocyte activation by bath application of t-ACPD (*Figure 1F*) dilatated arterioles in other chemoreceptor regions including the cNTS and ROb. This suggests that $CO_2/H^+$-dependent regulation of vascular tone and roles of astrocytes in this process are fundamentally different in the RTN compared to other chemoreceptor regions. Considering vasoconstriction and dilation are expected to increase and decrease tissue $CO_2/H^+$ levels respectively, we propose that $CO_2/H^+$ constriction in the RTN supports the drive to breathe by ensuring tissue acidification is maintained to stimulate chemoreceptors in this region, whereas $CO_2/H^+$ dilation in the cNTS and ROb provides a means of buffering tissue $H^+$ and chemoreceptor activity to maintain respiratory stability. Conversely, if $CO_2/H^+$ were to cause vasoconstriction across multiple chemoreceptor regions simultaneously, this may exaggerate the ventilatory response to $CO_2$ to the point of causing unstable periodic breathing (*Cherniack and Longobardo, 2006*). Indeed, this is thought to be the cause of unstable breathing in patients with global deficits in $CO_2/H^+$ vascular reactivity due to congestive heart failure or cerebrovascular disease (*Cherniack et al., 2005*; *Cherniack and Longobardo, 2006*). Consistent with this, we show that disruption of $CO_2/H^+$ dilatation in the cNTS and ROb by application of the vasoconstrictor U46619 while leaving RTN $CO_2/H^+$ constriction unperturbed, increased chemoreceptor drive as evidenced by a decrease in the $CO_2/H^+$ apneic threshold and caused unstable periodic breathing during exposure to higher levels of $CO_2/H^+$ (*Figure 3*). However, further work is needed to understand whether and how differential $CO_2/H^+$ vascular reactivity in these chemoreceptor regions contributes to breathing problems in disease states.

## Materials and methods

### Key resources table

| Reagent type (species) or resource | Designation | Source or reference | Identifiers | Additional information |
|---|---|---|---|---|
| Strain, strain background (*M. musculus*, Tie2-Cre, C57BL6/J background) | B6.Cg-Tg(Tek-cre) 1Ywa/J | Jackson Laboratories | RRID:IMSR_JAX:008863 | |
| Strain, strain background (*M. musculus*, smMHC-Cre/eGFP, C57BL6/J background) | B6.Cg-Tg(Myh11-cre, -EGFP)2Mik/J *Mus musculus* | Jackson Laboratories | RRID:IMSR_JAX:007742 | |

*Continued on next page*

*Continued*

| Reagent type (species) or resource | Designation | Source or reference | Identifiers | Additional information |
|---|---|---|---|---|
| Strain, strain background (aSM22-Cre, C57BL6/J background) | B6.Cg-Tg(Tagln-cre)1Her/J *Mus musculus* | Jackson Laboratories | RRID:IMSR_JAX:017491 | |
| Strain, strain background (TdTomato reporter Ai14, C57BL6/J background) | B6.Cg-Gt(ROSA)26Sor<sup>tm9(CAG-tdTomato)Hze</sup>/J *Mus musculus* | Jackson Laboratories | RRID:IMSR_JAX:007909 | |
| Strain, strain background (P2ry floxed, C57BL6/J background) | P2ry2<sup>f/f</sup> *Mus musculus* | PMID:27856454 | | Gifted by Dr. Cheike Seye (Indiana Univ.) |
| Transfected construct (*M. musculus*) | AAV2-Myh11p-eGFP-2A-mP2ry2 | This paper | | Custom AAV product from Vector Biolabs |
| Transfected construct (*M. musculus*) | AAV2-Myh11p-eGFP | This paper | | Custom AAV product from Vector Biolabs |
| Genetic reagent (TaKaRa TaqTM DNA Polymerase) | Taq polymerase | Promega | R001A | |
| Antibody | (mouse monoclonal) anti-α smooth muscle actin | Sigma | A5228 | (1:100 dilution) |
| Sequence-based reagent | Taqman probe *Gapdh* | ThermoFisher | Mm99999915_g1 | |
| Sequence-based reagent | Taqman probe *Rbfox3* | ThermoFisher | Mm01248771_m1 | |
| Sequence-based reagent | Taqman probe *Aldh1l1* | ThermoFisher | Mm03048957_m1 | |
| Sequence-based reagent | Taqman probe *Acta2* | ThermoFisher | Mm00725412_s1 | |
| Sequence-based reagent | Taqman probe *Flt1* | ThermoFisher | Mm00438980_m1 | |
| Sequence-based reagent | Taqman probe *P2rx1* | ThermoFisher | Mm00435460_m1 | |
| Sequence-based reagent | Taqman probe *P2rx2* | ThermoFisher | Mm00462952_m1 | |
| Sequence-based reagent | Taqman probe *P2rx3* | ThermoFisher | Mm00523699_m1 | |
| Sequence-based reagent | Taqman probe *P2rx4* | ThermoFisher | Mm00501787_m1 | |
| Sequence-based reagent | Taqman probe *P2rx5* | ThermoFisher | Mm00473677_m1 | |
| Sequence-based reagent | Taqman probe *P2rx6* | ThermoFisher | Mm00440591_m1 | |
| Sequence-based reagent | Taqman probe *P2rx7* | ThermoFisher | Mm01199500_m1 | |
| Sequence-based reagent | Taqman probe *P2ry1* | ThermoFisher | Mm02619947_s1 | |
| Sequence-based reagent | Taqman probe *P2ry2* | ThermoFisher | Mm02619978_s1 | |
| Sequence-based reagent | Taqman probe *P2ry4* | ThermoFisher | Mm00445136_s1 | |
| Sequence-based reagent | Taqman probe *P2ry6* | ThermoFisher | Mm02620937_s1 | |

*Continued on next page*

*Continued*

| Reagent type (species) or resource | Designation | Source or reference | Identifiers | Additional information |
|---|---|---|---|---|
| Sequence-based reagent | Taqman probe *P2ry12* | ThermoFisher | Mm01950543_s1 | |
| Sequence-based reagent | Taqman probe *P2ry13* | ThermoFisher | Mm01950543_s1 | |
| Sequence-based reagent | Taqman probe *P2ry14* | ThermoFisher | Mm01952477_s1 | |
| Chemical compound, drug | Papain | Sigma | P4762 | |
| Chemical compound, drug | 1,4-Dithioerythritol | Sigma | D8255 | |
| Chemical compound, drug | Collagenase IV | ThermoFisher | 1714019 | |
| Chemical compound, drug | Bovine Serum Albumin | Sigma | A2153 | |
| Chemical compound, drug | Trypsin Inhibitor | Sigma | T9253 | |
| Chemical compound, drug | U46619 | Tocris | 1932 | |
| Chemical compound, drug | Trans-ACPD | Tocris | 0187 | |
| Chemical compound, drug | AR-C 118925XX | Tocris | 4890 | |
| Chemical compound, drug | PSB 1114 | Tocris | 4333 | |
| Chemical compound, drug | POM 1 | Tocris | 2689 | |
| Chemical compound, drug | 1,3-dimethyl-8-phenyl-xantine (8-PT) | Sigma | P2278 | |
| Chemical compound, drug | Prostaglandin E2 | Cayman Chemical | 14010 | |
| Chemical compound, drug | L-798, 106 | Cayman Chemical | 11129 | |
| Chemical compound, drug | Papaverine | Sigma | P3510 | |
| Chemical compound, drug | Diltiazem | Tocris | 0685 | |
| Commercial assay, kit | Taqman Gene Expression Cells-to-CT kit | ThermoFisher | AM1728 | |
| Commercial assay, kit | Taqman Fast Advanced Master Mix | ThermoFisher | 4444557 | |
| Software, algorithm | ImageJ | NIH | RRID:SCR_003070 | Version 2.0.0 |
| Software, algorithm | Macro (for vessel analysis) | PMID:28387198 | | |
| Software, algorithm | Ponemah | DSI | RRID:SCR_017107 | Version 5.23 |
| Software, algorithm | QuantStudio Design and Analysis | ThermoFisher | RRID:SCR_018712 | Version 1.5.1 |
| Software, algorithm | Prism 7 | GraphPad | RRID:SCR_011323 | Version 7.03 |
| Other | (Griffonia Simplicifolia Lectin I) isolectin B4, Dylight 594 | Vector Laboratories | SL-1207-.5 | (6 µg/mL dilution) |

## Animals

All procedures were performed in accordance with National Institutes of Health and University of Connecticut Animal Care and Use Guidelines. All experiments used mixed sex C57BL6/J animals at least 3 weeks of age that housed in a 12:12 light dark cycle with normal chow ad libitum. The following transgenic Cre mouse lines were used to perform single cell isolation, FACS, and pooled qPCR: $Myh11^{Cre/eGFP}$ (Jax Stock #: 007742) and $Tek^{Cre}$::TdTomato (Jax Stock #: 008863, 007909), and $Tagln^{Cre}$::TdTomato (Jax Stock #: 017491, 007909). $Tagln^{Cre}$ and $Myh11^{Cre/eGFP}$ mice were crossed with $P2ry2$ floxed mice (gifted by Dr. Cheike Seye at Indiana University) to generate double floxed smooth muscle-specific P2Y$_2$ cKO mice for in vitro and in vivo experimentation.

## In vitro arteriole slice recordings

### Brainstem and cortical slice preparation

Animals (P21 and older) were decapitated under isoflurane anesthesia and transverse brainstem slices (150 μm) were prepared using a vibratome in ice-cold substituted artificial cerebrospinal fluid (aCSF) solution containing (in mM): 130 NaCl, 3 KCl, 2 MgCl$_2$, 2 CaCl$_2$, 1.25 NaH$_2$PO$_4$, 26 NaHCO$_3$, 10 glucose. 0.4 mM L-ascorbic acid was added into aCSF while slicing. Slices were incubated for 30 min at 37°C and subsequently at room temperature in aCSF, equilibrated with a 5% CO$_2$-95% O$_2$ gas mixture. Prior to imaging, each slice was incubated for 35 min with 6 mg/mL DyLight 594 Isolectin B4 conjugate (Vector Labs) to label vascular endothelium as previously described (*Hawkins et al., 2017*).

### Imaging arterioles in vitro

Individual brain slices containing either the caudal NTS (cNTS), raphe obscurus (ROb), RTN, or somatosensory cortex were transferred to a Plexiglas recording chamber mounted on a fixed-stage upright fluorescent microscope (Zeiss Axioskop FS) and perfused with 37°C aCSF bubbled with a 5% CO$_2$-21% O$_2$ (balance N$_2$). Hypercapnic solutions were made by allowing aCSF to equilibrate with a 15% CO$_2$-21% O$_2$ (balance N$_2$). Arterioles were identified based on clear evidence of vasomotion under IR-DIC microscopy and bulky fluorescent labeling that indicates a thick layer of tightly wrapped smooth muscle surrounding the vessel lumen. Precapillary arterioles (as indicated by more sporadic, less uniform smooth muscle cell layer and thinner luminal diameter) were excluded from experimentation. All arterioles selected for experimentation had a luminal diameter between 8–50 μm. It should be noted that most, if not all, of the cNTS arterioles were small (average diameter was 9 μm) and ROb vasculature did not vary more than 10 μm from the midline of the slice. RTN vessels were located within 200 μm of the ventral surface and below the caudal end of the facial motor nucleus. Neocortical vessels were located in layers 1–5.

For an experiment, fluorescent images were acquired at a rate of 1 frame every 20 s using a 40X water objective lens, a Clara CCD Andor camera, and the NIS Advanced Research software suite (Nikon). To induce a partially constricted state in arterioles, we continuously bath applied 125 nM of a thromboxane A2 receptor agonist (U46619). U46619 has been previously shown to induce a 20–30% vasoconstriction at a concentration of 125 nM, enabling the vessel to either dilate or constrict further under experimental pharmacological studies (*Girouard et al., 2010*). As previously described (*Hawkins et al., 2017*), we assessed arteriole viability at the end of each experiment by inducing a large vasoconstriction with a 60 mM K$^+$ solution and then a large vasodilation with a Ca$^{+2}$-free solution containing EGTA (5 mM), a phosphodiesterase inhibitor (papaverine, 200 mM), and an L-type Ca$^{+2}$ channel blocker (diltiazem, 50 mM). Only one vessel was recorded per experiment and slice. Any vessel that did not respond to these solutions were excluded in data analysis. The list of all drugs and concentrations used for in vitro and in vivo experimentation are detailed in Key Resource Table and text where appropriate.

### Image analysis

Vessel diameter was determined using ImageJ. All images were calibrated, and pixel distance was converted to millimeters. Data files underwent StackReg (Biomedical Imaging Group) to stack each image over time and then three linear region of interests (ROI) were drawn orthogonal to the vessel. A macro (available at https://github.com/omsai/blood-vessel-diameter [Nanda, 2017]) was used to

determine peak to peak distance using fluorescence intensity profile plots for all slices of the data file.

## Single-cell isolation and qRT-PCR

At least three and no greater than eight, animals (postnatal days P21-P40) were used of either of the following genotypes: Myh11[Cre/eGFP], Tek[Cre]::TdTomato, Tagln[Cre]::TdTomato, or Tagln[Cre]::TdTomato::P2ry2[fl/fl]. Animals were euthanized under isoflurane anesthesia and brainstem slices were prepared using a vibratome in ice cold, high sucrose slicing solution containing (in mM): 87 NaCl, 75 sucrose, 25 glucose, 25 NaHCO$_3$, 1.25 NaH$_2$PO$_4$, 2.5 KCl, 7.5 MgCl$_2$, 0.5 mM CaCl$_2$ and 5 L-ascorbic acid. Slicing solution was equilibrated with a 5% CO$_2$-95% O$_2$ gas mixture before use. Transverse brainstem slices (400 μm thick) were prepared and then immediately enzymatically treated at 34°C with an initial incubation in a papain (6 mg/mL, Sigma) and dithioerythritol (10 mg/mL, Thermo-Fisher) mixture for 30 min in sucrose dissociation solution containing (in mM): 185 sucrose, 10 glucose, 30 Na$_2$SO$_4$, 2 K$_2$SO$_4$, 10 HEPES, 0.5 CaCl$_2$, 6 MgCl$_2$, 5 L-ascorbic acid, pH 7.4, 320 mOsm, followed by a collagenase IV (10 mg/mL, ThermoFisher) enzyme treatment for 6 min. After enzyme incubation, slices were washed three times in cold dissociation solution and then transferred to an enzyme inhibitor mix containing trypsin inhibitor (10 mg/mL, Sigma), bovine serum albumin (BSA, 10 mg/mL, Sigma), and sodium nitroprusside (SNP, 2 mg/mL, ThermoFisher) in cold sucrose dissociation solution. Shortly thereafter, slices were transferred to a glass Petri dish on ice. Using a plastic transfer pipette and a 15-blade scalpel, each region of interest was microdissected out of the slices and manually separated into sterile microcentrifuge tubes. The control samples were made up of two slices: one slice with the NTS removed and the other with the ROb and RTN removed. Once microdissection is completed, the tissue chunks were warmed to 34°C for 10 min before trituration. A single-cell suspension was achieved by trituration using a 25G and 30G needle sequentially, attached to a 3 mL syringe. Samples were triturated for an average of 5 min. Immediately after, the samples were placed back on ice and filtered through a 30-μm filter (Miltenyi Biotech) into round bottom polystyrene tube for fluorescence-activated cell sorting (FACS).

### Florescence-activated cell sorting (FACS)

All cell types of interest were sorted on a BD FACSAria II Cell Sorter (UConn COR$^2$E Facility, Storrs, CT) equipped with 407 nm, 488 nm, and 607 nm excitation lasers. Five minutes before sorting, 5 μL of 100 ng/mL DAPI was added to each sample. Cells were gated based on scatter (forward and side), for singlets, and for absence of DAPI. Finally, cells were gated either to TdTomato or GFP fluorescence and sorted by four-way purity into a sterile 96-well plate containing 5 μL of sterile PBS per sample. Between 100 and 500 cells were sorted per sample in any experiment and were processed immediately following FACS. To control for RNA contamination in FACS droplets, allophycocyanin (APC) beads were sorted based on green fluorescence and were treated alongside experimental samples (data not shown). See *Figure 1—figure supplement 1* for FACS gating parameters.

### Pooled cell qRT-PCR

A lysis reaction followed by reverse transcription was performed using the kit Taqman Gene Expression Cells-to-CT Kit (ThermoFisher) with 'Lysis Solution' followed by the 'Stop Solution' at room temperature, and then a reverse transcription with the 'RT Buffer', 'RT Enzyme Mix', and lysed RNA at 37°C for an hour. Following reverse transcription, cDNA was pre-amplified by adding 2 μL of cDNA from each sample to 8 μL of preamp master mix [5 μL TaKaRa premix Taq polymerase (Clontech), 2.5 μL 0.2X Taqman pooled probe, 0.5 μL H$_2$O] and thermocycled at 95°C for 3 min, 55°C for 2 min, 72°C for 2 min, then 95°C for 15 s, 60°C for 2 min, 72°C for 2 min for 16–20 cycles, and then a final 10°C hold. Amplified cDNA was then diluted 2:100 in RNase-free H$_2$O. Each qPCR assay contained the following reagents: 0.5 uL 20X Taqman probe, 2.5 μL RNase-free H$_2$O, 5 μL Gene Expression Master Mix or Fast Advanced Master Mix (ThermoFisher), and 2 L diluted pre-amplified cDNA. qPCR reactions were performed in triplicate for each Taqman assay of interest on a QuantStudio 3 Real Time PCR Machine (ThermoFisher).

### qRT-PCR data analysis

All three technical replicates were averaged to create one raw Ct values per Taqman assay. As a control, all samples were subject to cell markers of various cell types to confirm specificity: *Rbfox3* for neurons, *Aldh1l1* for astrocytes, *Acta2* for smooth muscle cells, and *Flt1* for endothelial cells (*Figure 1A–B*, *Figure 1—source data 1*). Any assay that did not give a discrete Ct was given a Ct value of 40 for analysis. Fold change was determined by using the equation $2^{(-\Delta\Delta Ct)}$. *Gapdh* was used as a sample-dependent internal control. Fold changes for all runs of each cell type were averaged; the $\log_2$ of the fold change was calculated, analyzed, and plotted on bar graphs.

## In vivo anesthetized preparation

Animal use was in accordance with guidelines approved by the University of São Paulo Animal Care and Use Committee. A total of 26 adult male C57BL6 (25–28 g) were used for in vivo experiments. General anesthesia was induced with 5% isoflurane in 100% O2. A tracheostomy was made, and the isoflurane concentration was reduced to 1.4–1.5% until the end of surgery. The carotid artery was cannulated (polyethylene tubing, 0.6 mm o.d., 0.3 mm i.d., Scientific Commodities) for measurement of arterial pressure (AP). The jugular vein was cannulated for administration of fluids and drugs. Rats were placed supine onto a stereotaxic apparatus (Type 1760; Harvard Apparatus) on a heating pad and core body temperature was maintained at a minimum of 36.5°C via a thermocouple. The trachea was cannulated. Respiratory flow was monitored via a flow head connected to a transducer (GM Instruments) and $CO_2$ via a capnograph (CWE, Inc,) connected to the tracheal tube. Paired EMG wire electrodes (AM-System) were inserted into the external intercostal muscle to record respiratory-related activity. After the anterior neck muscles were removed, a basio-occipital craniotomy exposed the ventral medullary surface, and the dura was resected. After bilateral vagotomy, the exposed tissue around the neck and the mylohyoid muscle was covered with mineral oil to prevent drying. Baseline parameters were allowed to stabilize for 30 min prior to recording.

### In vivo recordings of physiological variables

Mean arterial pressure (MAP), external intercostal muscle activity ($Int_{EMG}$) and end-expiratory $CO_2$ ($etCO_2$) were digitized with a micro1401 (Cambridge Electronic Design), stored on a computer, and processed off-line with version 7 of Spike two software (Cambridge Electronic Design). Integrated intercostal activity ($\int Int_{EMG}$) was collected after rectifying and smoothing ($\tau = 0.03$) the original signal, which was acquired with a 300–3000 Hz bandpass filter. Noise was subtracted from the recordings prior to performing any calculations of evoked changes in EMG. A direct physiological comparison of the absolute level of EMG activity across nerves is not possible because of non-physiological factors (e.g. muscle electrode contact, size of muscle bundle) and the ambiguity in interpreting how a given increase in voltage in one EMG relates to an increase in voltage in another EMG. Thus, muscle activity was defined according to its baseline physiological state, just prior to their activation. The baseline activity was normalized to 100%, and the percent change was used to compare the magnitude of increases or decreases across muscle from those physiological baselines.

### In vivo experimental protocol

Each in vivo experiment began by testing responses to hypercapnia by adding $CO_2$ to the breathing air supplied by artificial ventilation. The addition of $CO_2$ was monitored to reach a maximum end-expiratory $CO_2$ between 9% and 10%, which corresponds with an estimated arterial pH of 7.2 based on the following equation: $pHa = 7.955 - 0.7215 \times \log10\ (EtCO_2)$. This maximum end-expiratory $CO_2$ was maintained for 5 min and then replaced by 100% $O_2$.

To determine whether local regulation of vascular tone in the region of the RTN contributes to the $CO_2/H^+$-dependent drive to breathe, we made injections of saline, PSB1114 (100 µM), AR-C118925 (1 mM) or U46619 (1 µM) while monitoring $Int_{EMG}$ amplitude and frequency. These drugs were diluted with sterile saline (pH 7.4) and applied using single-barrel glass pipettes (tip diameter of 25 µm) connected to a pressure injector (Picospritzer III, Parker Hannifin Corp, Cleveland, OH). For each injection, we delivered a volume of 30 nL over a period of 5 s. Injections in the VMS region were placed 1.9 mm lateral from the basilar artery, 0.9 mm rostral from the most rostral hypoglossal nerve rootlet, and at the VMS. The second injection was made 1–2 min later at the same level on the contralateral side. For injections located at the cNTS or ROb, we used the following coordinates: (a)

cNTS: 0.2–0.3 mm rostral to the *calamus scriptorius*, 0.3 mm lateral to midline, and 0.3 mm below the dorsal surface of the brainstem and ROb: 1.2–1.3 mm caudal to the parietal-occipital suture, in the midline, and 5–5.3 mm below the cerebelar surface.

A cranial optical window was prepared using standard protocols. For the VMS, the anterior neck muscles were removed, a basio-occipital craniotomy exposed the ventral medullary surface, and the dura was resected. Pial vessels in the VMS were located 1.9 mm lateral from the basilar artery and 0.9 mm rostral to the most rostral portion of the hypoglossal nerve rootlet. The surface of the VMS was cleaned with buffer containing (in mmol/L) the following: 135 NaCl, 5.4 KCl, 1 $MgCl_2$, 1.8 $CaCl_2$, and 5 HEPES, pH 7.3., and a chamber (home-made 1.1-cm-diameter plastic ring was glued with dental acrylic cement attached to a baseplate). The chamber was sealed with a circular glass coverslip (#1943–00005, Bellco). The baseplate was affixed to the Digital Camera (Sony, DCR-DVD3-5) and a light microscope was used for vessel imaging (x40 magnification).

## In vivo pial vessel imaging

### Animal preparation

Mixed sex adult mice (>6 weeks of age) were briefly anesthetized with isoflurane (1–3%) followed with an IP injection of 300–500 mg urethane to ensure a deep anesthetic state. The animal was then fixed to stereotaxic ear bars with the ventral side of the animal facing up. The trachea of the animal was then cannulated with a 18G needle to provide 1.5 mL ventilations at a rate of 150 breaths per minute of 5% $CO_2$-21% $O_2$ (balance $N_2$) via artificial ventilator (Kent Scientific). After cannulation, deep neck muscles were resected, and the cranial-pharyngeal canal and dura mater of the animal was removed. A digital camera was placed over the ventral surface and focused on the pial vasculature. Animals that did not have well-perfused pial vessels were not included in experimentation. Animals that were excessively bleeding from the surgical site were sacrificed and not included in any analyses.

### In vivo image analysis

Three linear ROIs were drawn over the pial vasculature we recognize as supplying blood flow to the RTN, similar to the in vitro methods described above. Bright field imaging was analyzed using a macro in ImageJ. Briefly, the macro measured the drop and rise of the local maxima and minima, correlating to vessel boundaries. These boundaries were then used to make a linear measurement between the two points which was vessel diameter.

## RTN viral injections

Adult mice (>20 grams) were anesthetized with 3% isoflurane. The right cheek of the animal was shaved, and an incision was made to expose the right marginal mandibular branch of the facial nerve. The animals were then placed in a stereotaxic frame and a bipolar stimulating electrode was placed directly adjacent to the nerve. Animals were maintained on 1.5% isoflurane for the remainder of the surgery. An incision was made to expose the skull and two 1.5 mm holes were drilled left and right of the posterior fontanelle, caudal of the lambdoidal suture. The facial nerve was stimulated using a bipolar stimulating electrode to evoke antidromic field potentials within the facial motor nucleus. In this way, the facial nucleus on the right side of the animal was mapped in the X, Y, and Z direction using a quartz recording electrode.

The viral vector was loaded into a 1.2 mm internal diameter borosilicate glass pipette on a Nanoject III system (Drummond Scientific). Virus was injected at least −0.02 mm ventral to the Z coordinates of the facial nucleus, to ensure injection into the RTN. These same coordinates were used for the left side of the animal. In all mice, incisions were closed with nylon sutures and surgical cyanoacrylate adhesive. Mice were placed on a heated pad until consciousness was regained. Meloxicam was administered 24 and 48 hr postoperatively.

## Telemetry transmitter placement

Adult mice (>30 grams) were anesthetized with an induction dose of 3% isoflurane and placed into a sterile field. HD-X11 transmitters (Data Sciences International; DSI) were placed and fixed intraperitoneally with non-absorbable nylon sutures, followed by ECG lead placement on the left and right pectoralis on top of the rib cage. Finally, a pressure catheter was placed into the left carotid artery and

threaded through to the aortic arch for continuous blood pressure measurements. Animals were closed with nylon sutures and placed on a heated pad u conscious. Mice were continuously watched for the next three days for any post-operative pain complications while meloxicam was administered 24 and 48 hr postoperatively. Mice were not used for any experimentation until post-operative day 7.

## Unrestrained whole-body plethysmography

Respiratory activity was measured using a whole-body plethysmograph system (Data Scientific International; DSI), utilizing animal chamber maintained at room temperature and ventilated with air (1.3 L/min) using a small animal bias flow generator. Mice were individually placed into a chamber and allowed 1 hr to acclimate prior to the start of an experiment. Respiratory activity was recorded using Ponemah 5.32 software (DSI) for a period of 15 min in room air followed by exposure to graded increases in $CO_2$ from 0% to 7% $CO_2$ (balance $O_2$). In separate experiments, we characterized the ventilatory response to 10% $O_2$ (balance $N_2$). Parameters of interests include respiratory frequency ($F_R$, breaths per minute), tidal volume ($V_T$, measured in mL; normalized to body weight and corrected to account for chamber and animal temperature, humidity, and atmospheric pressure), and minute ventilation ($V_E$, mL/min/g). A 20 s period of relative quiescence after 5 min of exposure to each condition was selected for analysis. All experiments were performed between 9 a.m. and 6 p.m. to minimize potential circadian effects.

## Comprehensive lab animal monitoring (CLAMS)

Metabolic monitoring ($VCO_2$, $VO_2$) was performed using comprehensive lab animal monitoring systems (CLAMS, Columbus Instruments). In short, mice were individually housed on a 12:12 light dark cycle in plastic cages with a running wheel, regular bedding, and regular chow for 1 week before experimentation. Three days before the metabolic experiment, each animal was placed in the CLAMS housing cage with metered water and waste collection. Mice had 2 days to acclimate to the metabolic chamber; on the third day, all results were recorded for a continuous 24 hr period. After data collection, all raw results were exported and averaged out per hour. Then, light and dark periods were determined and averaged per animal for statistical analysis. Both sexes were equally represented in the data set.

## Blood gas analysis

Arterial blood gasses were collected from adult mice 6 weeks of age and older (>30 grams). A RAPI-DLab 348 blood gas analyzer (Siemens) was used for all blood gas analysis; all calibrations, QC, and use were performed as indicated by the manufacturer. Animals were anesthetized with an induction dose of 3% isoflurane and then quickly switched to 1% isoflurane for the remainder of arterial blood collection. This level of isoflurane minimally affects blood gases (*Constantinides et al., 2011*; *Loeven et al., 2018*). The left carotid artery was exposed and quickly cannulated to allow for arterial blood to be collected and analyzed by the blood gas analyzer; no more than 5 s was spent between blood collection and analysis on the blood gas analyzer.

## Immunohistochemistry

Adult mice were transcardially perfused with 20 mL of room temperature phosphate buffered saline (PBS, pH 7.4) followed by 20 mL of chilled 4% paraformaldehyde (pH 7.4). The brainstem was then removed from the animal and 150 um slices were made using a Zeiss VT100S vibratome. Slices were then incubated in a 0.5% Triton-X/PBS solution for 45 min to permeabilize the tissue. The slices remained in a 0.1% Triton-X/10% Fetal Bovine Serum (FBS, ThermoFisher)/PBS solution for a 12 hr primary antibody incubation of anti-mouse alpha smooth muscle actin (Sigma). The tissue was then washed three times in 0.1% Triton-X/10% FBS/PBS solution; the secondary antibody was incubated with the tissue after the third wash for 2 hr (donkey anti-mouse Alexa Fluor 647, ThermoFisher). The tissue was then washed three times in PBS before mounting on precleaned cover slides with Prolong Diamond with DAPI (ThermoFisher). Imaging of brain slices was achieved with a Leica SP8 confocal microscope.

## Statistics

Data are reported as mean ± SE. Power analysis was used to determine sample size, all data sets were tested for normality using Shapiro-Wilk test, and comparisons were made using t-test, one-way or two-way ANOVA (parametric or non-parametric) followed by multiple comparison tests as appropriate. The specific test used for each comparison is reported in the figure legend and all relevant values used for statistical analysis are included in the results section.

## Acknowledgements

The authors thank Drs. Douglas Bayliss (Univ. Virginia) and Akiko Nishiyama (Univ. Connecticut) for their comments on the manuscript. This work was supported by funds from the National Institutes of Health Grants HL104101 (DKM), HL137094 (DKM), NS099887 (DKM) R01NS110656 (MTN), R35HL140027 (MTN) and F31HL142227 (CMC). Additional funds were also provided by the American Heart Association (17SDG33670237 and 19IPLOI34660108 to TAL), the São Paulo Research Foundation (FAPESP; grants: 2019/01236-4 to ACT; 2015/23376-1 to TSM) and the Conselho Nacional de Desenvolvimento Científico e Tecnológico (CNPq; grant: 408647/2018–3 to ACT). CNPq fellowships were awarded to ACT (302288/2019-8) and to TSM (302334/2019-0). This study was also financed by the Coordenação de Aperfeiçoamento de Pessoal de Nível Superior - Brasil (CAPES) - Financial Code 001 and by Serrapilheira Institute (Serra-1812-26431 to ACT). This work was also supported by Fondation Leducq (Transatlantic Network of Excellence on the Pathogenesis of Small Vessel Disease of the Brain) (MTN), the European Union (Horizon 2020 Research and Innovation Programme SVDs@target under the grant agreement n° 666881) (MTN) and the Henry M Jackson Foundation for the Advancement of Military Medicine, HU0001-18-2-0016 (MTN).

## Additional information

### Competing interests

Mark T Nelson: Reviewing editor, *eLife*. The other authors declare that no competing interests exist.

### Funding

| Funder | Grant reference number | Author |
| --- | --- | --- |
| National Institutes of Health | HL104101 | Daniel K Mulkey |
| National Institutes of Health | HL137094 | Daniel K Mulkey |
| National Institutes of Health | NS099887 | Daniel K Mulkey |
| National Institutes of Health | NS110656 | Mark T Nelson |
| National Institutes of Health | HL140027 | Mark T Nelson |
| National Institutes of Health | HL142227 | Colin M Cleary |
| American Heart Association | 17SDG33670237 | Thomas A Longden |
| American Heart Association | 19IPLOI34660108 | Thomas A Longden |
| São Paulo Research Foundation | 2016/23281-3 | Ana C Takakura |
| São Paulo Research Foundation | 2015/23376-1 | Thiago S Moreira |
| Conselho Nacional de Desenvolvimento Científico e Tecnológico | 408647/2018-3 | Ana C Takakura |
| Conselho Nacional de Desenvolvimento Científico e Tecnológico | 301219/2016-8 | Ana C Takakura |
| Conselho Nacional de Desenvolvimento Científico e Tecnológico | 301904/2015-4 | Thiago S Moreira |

| Fondation Leducq | | Mark T Nelson |
| Horizon 2020 - Research and Innovation Framework Programme | | Mark T Nelson |
| Henry M. Jackson Foundation | HU0001-18-2-001 | Mark T Nelson |

The funders had no role in study design, data collection and interpretation, or the decision to submit the work for publication.

## Author contributions
Colin M Cleary, Data curation, Formal analysis, Writing - review and editing; Thiago S Moreira, Ana C Takakura, Data curation, Formal analysis, Funding acquisition, Investigation, Project administration, Writing - review and editing; Mark T Nelson, Conceptualization, Funding acquisition, Project administration, Writing - review and editing; Thomas A Longden, Conceptualization, Funding acquisition, Investigation, Writing - review and editing; Daniel K Mulkey, Conceptualization, Funding acquisition, Investigation, Writing - original draft, Project administration, Writing - review and editing

## Author ORCIDs
Colin M Cleary  https://orcid.org/0000-0003-0305-1324
Thiago S Moreira  http://orcid.org/0000-0002-9789-8296
Mark T Nelson  http://orcid.org/0000-0002-6608-8784
Daniel K Mulkey  https://orcid.org/0000-0002-7040-3927

## Ethics
Animal experimentation: All procedures were performed in accordance with National Institutes of Health and University of Connecticut Animal Care and Use Guidelines as described in protocols A19-048 and A20-016.

## Decision letter and Author response
Decision letter https://doi.org/10.7554/eLife.59499.sa1
Author response https://doi.org/10.7554/eLife.59499.sa2

# Additional files

## Supplementary files
• Transparent reporting form

## Data availability
Source data files are included for all data sets that do not have individual points on summary figures.

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
