## [Decision Letter]

Thank you for submitting your article "Vascular control of the CO_2_/H^+^ dependent drive to breathe" for consideration by *eLife*. Your article has been reviewed by three peer reviewers, including Jeffrey C Smith as the Reviewing Editor and Reviewer #3, and the evaluation has been overseen by Huda Zoghbi as the Senior Editor. The following individual involved in review of your submission has agreed to reveal their identity: Adrianne G Huxtable (Reviewer #1).

The reviewers have discussed the reviews with one another and the Reviewing Editor has drafted this decision to help you prepare a revised submission.

As the editors have judged that your manuscript is of interest, but as described below that additional data are required before it is published, we would like to draw your attention to changes in our revision policy that we have made in response to COVID-19 (https://elifesciences.org/articles/57162). First, because many researchers have temporarily lost access to the labs, we will give authors as much time as they need to submit revised manuscripts. We are also offering, if you choose, to post the manuscript to bioRxiv (if it is not already there) along with this decision letter and a formal designation that the manuscript is "in revision at *eLife*". Please let us know if you would like to pursue this option. (If your work is more suitable for medRxiv, you will need to post the preprint yourself, as the mechanisms for us to do so are still in development.)

Summary:

This paper is an important extension of the authors' previous publication in *eLife* (Hawkins et al., 2017) that presented novel data suggesting that CO_2_/H^+^-mediated vasoconstriction in the brainstem retrotrapezoid nucleus (RTN) supports chemoreception by a purinergic-dependent mechanism. Here the investigators provide new data indicating that CO_2_/H^+^ dilates arterioles in other chemoreceptor regions (cNTS, raphe obscurus- ROb), thus suggesting that the CO_2_/H^+^ vascular reactivity in the RTN is unique compared to some other brain regions. The investigators significantly advance their previous work by applying a number of new experimental approaches to provide evidence that P2Y_2_ receptors in RTN vascular smooth muscle cells are responsible for the purinergic mechanism mediating the vascular reactivity and specifically contribute to RTN chemosensitivity. Importantly, pharmacological blockade or genetic deletion of P2Y_2_ from smooth muscle cells blunted the in vivo ventilatory response to CO_2_, and virally-driven re-expression of P2Y_2_ receptors in RTN smooth muscle cells rescued the ventilatory response to CO_2_, suggesting that these receptors are required for the normal ventilatory response to CO_2_. New pharmacological evidence is also presented that activation of RTN astrocytes is involved in purinergic signaling driving the RTN vasomotor responses. Overall these results advance the concept that specialized vasoreactivity to CO_2_/H^+^ in the RTN contributes to respiratory chemoreception.

Essential revisions:

1) Although authors are given leeway in the format of a Research Advance, this paper would benefit from more structure including delineation of Introduction, Results, and Discussion sections. The manuscript would be substantially improved in particular by including a more thorough, dedicated Discussion section with explicit elaboration on limitations of their experimental methods and conclusions, and including discussion of how the important P2Y_2_ receptor knockout and re-expression experiments represent a fundamental advance considering that the authors had already implicated (although not completely established) these receptors in their previous publication.

2) Presentation of the RT-PCR data of purinergic receptor expression profiles can be improved, particularly by providing a more convincing validation of this data such as giving supplementary data of raw numbers for GAPDH levels across areas to prove that GAPDH actually is a valid reference. The authors could also use 3-4 such genes as many investigators do for expression profile calibration. The reviewers note that for the argument it is not necessarily that important how the levels of receptors look in relation to a house keeping gene, but whether P2Y_2_ is the only receptor which is relatively highly expressed in RTN smooth muscle cells compared to other regions. Looking at Figure 1B, it seems that relative to the two other areas, P2X1, P2X4 and P2Y14 are also much higher in RTN smooth muscle cells compared to NTS. The reviewers agree that an important aspect is the remarkably low expression of P2Y_2_ in endothelium which in theory should oppose constriction by possibly releasing NO.

3) Additional information on measurements of vascular diameters would be useful. Have the authors obtained measurements from multiple vessels at each time point in the chosen field(s) of view for individual experiments? If so, how do such measurements compare to the representative single vessel measurements for a given experiment presented in the figures? How many vessels per experiment are included in the group summary data? Please explain more completely why it was necessary to induce a 20-30% vasoconstriction by the thromboxane A2 receptor agonist before the measurements.

4) Some additional validation of the specificity of the AAV2 used for the P2Y_2_ re-expression experiments would be helpful since this is not a well characterized virus and may lead to receptor overexpression. Additional nice clear images with proper co-localization would be good to see and additional details about non-smooth muscle cell expression should be provided.

5) The experiments showing unstable breathing in vivo produced by injecting a thromboxane A2 receptor agonist vasoconstrictor (U46119) into the cNTS and ROb under conditions of mild hypercapnia (2-3% inspired CO_2_) are intriguing, but these experiments lack the proper control of U46119 injections into the cNTS and ROb under normocapnic conditions to determine if this alters blood pressure and produces breathing instabilities independent of any "gain-up" of RTN activity. It would also be of interest to know whether the authors have tested if larger instabilities occur with cNTS/ROb vasoconstriction at higher levels of hypercapnia.

---

## [Author Response]

Essential revisions:1) Although authors are given leeway in the format of a Research Advance, this paper would benefit from more structure including delineation of Introduction, Results, and Discussion sections. The manuscript would be substantially improved in particular by including a more thorough, dedicated Discussion section with explicit elaboration on limitations of their experimental methods and conclusions, and including discussion of how the important P2Y_2_ receptor knockout and re-expression experiments represent a fundamental advance considering that the authors had already implicated (although not completely established) these receptors in their previous publication.

We have added an expanded Discussion section that includes experimental limitations section. We also make clear how this work extends our previous findings. Specifically, our previous work showed in vitro that a non-specific blocker of P2Y_2_ and P2Y6 blunted RTN CO_2_/H^+^ vascular reactivity. We also showed in anesthetized rats that blocking all P2 receptors disrupted CO_2_/H^+^ vascular reactivity and the ventilatory response to CO_2_. The current study builds on this work by showing i) CO_2_/H^+^ vascular responses in the RTN are opposite to other chemoreceptor regions; ii) vascular endothelial cells and smooth muscle cells in the RTN exhibit a P2 receptor profile that differs from other chemoreceptor regions and favors constriction via P2Y_2_; iii) specific pharmacological manipulations of P2Y_2_ mimic and block CO_2_/H^+^ vascular reactivity in vitro and in anesthetized mice; and iv) conditional ablation of P2Y_2_ from smooth muscle cells blunted the ventilatory response to CO_2_ in awake mice, and re-expression of P2Y_2_ only in RTN smooth muscle cells rescued the ventilatory response to CO_2_. These results definitively establish P2Y_2_ receptors in RTN smooth muscle cells as important determinants of the drive to breathe.

2) Presentation of the RT-PCR data of purinergic receptor expression profiles can be improved, particularly by providing a more convincing validation of this data such as giving supplementary data of raw numbers for GAPDH levels across areas to prove that GAPDH actually is a valid reference. The authors could also use 3-4 such genes as many investigators do for expression profile calibration. The reviewers note that for the argument it is not necessarily that important how the levels of receptors look in relation to a house keeping gene, but whether P2Y_2_ is the only receptor which is relatively highly expressed in RTN smooth muscle cells compared to other regions. Looking at Figure 1B, it seems that relative to the two other areas, P2X1, P2X4 and P2Y14 are also much higher in RTN smooth muscle cells compared to NTS. The reviewers agree that an important aspect is the remarkably low expression of P2Y_2_ in endothelium which in theory should oppose constriction by possibly releasing NO.

We have added a supporting table (Figure 1—source data 1) that includes all raw Ct values for Gapdh and cell-type specific genes including Rbfox3 (neurons), Aldh1L1 (astrocytes), Acta2 (smooth muscle cells) and Flt1 (endothelial cells) for both endothelial and smooth muscle cell populations. The presence/absence of cell type specific genes was used to confirm that we obtained enriched populations of endothelial cells (Figure 1A) or smooth muscle cells (Figures 1B and 4A). Gapdh showed fairly consistent and repeatable Ct values across all regions of interest, and so was used an internal control for results shown in Figures 1A-B and 4A.

The first section of the Results has been updated to include a comparison of P2 transcript levels between the regions of interest while continuing to highlight the unusual expression profile of *P2ry2* in the RTN.

3) Additional information on measurements of vascular diameters would be useful. Have the authors obtained measurements from multiple vessels at each time point in the chosen field(s) of view for individual experiments? If so, how do such measurements compare to the representative single vessel measurements for a given experiment presented in the figures?

It was uncommon to find multiple vessels in focus within a field of view so for this work we zoomed in on just one vessel per experiment. For an experiment, we monitored vessel diameter over time at three locations along the vessel within the region of interest, and the average of those measurements was used to assess individual vascular responses.

In any case, we agree that it would be very interesting to understanding how CO_2_/H^+^ affects the larger vascular network within and near each chemoreceptor region. This is a direction we plan to pursue in the future.

How many vessels per experiment are included in the group summary data?

This information has been added to the text.

Please explain more completely why it was necessary to induce a 20-30% vasoconstriction by the thromboxane A2 receptor agonist before the measurements.

In intact tissue, blood flow subjects’ vascular smooth muscle to a degree of stretching that results in an autoregulatory increase in myogenic tone. The absence of blood flow in brain slices results in a loss of myogenic tone. In an effort to see vascular responses in either direction (dilation or constriction) we pre-constrict vessels ~30% from maximum dilation using a thromboxane A2 agonist U-46619. This approach is commonly used to study vascular regulation in the brain slice preparation (PMID: 18456724). It is important to note that RTN vascular responses to CO_2_/H^+^ and P2Y_2_ agonist/antagonist observed in vitro were also confirmed in vivo in the absence of U-46691. We have included this point in the limitations section of the Discussion.

4) Some additional validation of the specificity of the AAV2 used for the P2Y_2_ re-expression experiments would be helpful since this is not a well characterized virus and may lead to receptor overexpression.

Good suggestion. We characterized P2Y_2_ transcript in RTN, brainstem and cortical smooth muscle cells isolated from control mice (*Tagln*^Cre^::TdTomato), P2Y_2_ cKO mice (*Tagln*^Cre^::*P2ry2*^f/f^::TdTomato), and P2Y_2_ rescue mice (P2Y_2_ cKO animals that received bilateral RTN injections of AAV2-*Myh11*p-eGFP-2A-m*P2ry2*). P2Y_2_ transcript was not detected in smooth muscle cells from P2Y_2_ cKO mice. P2Y_2_ was also not detected in cortical smooth muscle cells from either genotype. Conversely, RTN and brainstem but not cortical smooth muscle cells from P2Y_2_ rescue mice show P2Y_2_ transcript levels in similar to control. These results have been added to Figure 4A.

Additional nice clear images with proper co-localization would be good to see and additional details about non-smooth muscle cell expression should be provided.

We have added a new supplementary figure (Figure 4—figure supplement 4) that provides several examples of viral transduced smooth muscle cells (eGFP+ and Acta2+). This figure also shows an example of small eGFP puncta in an Acta2-negative cell (Figure 4—figure supplement 4Aii) and examples of eGFP puncta not co-localized with DAPI. Because off-target eGFP labeling was relatively weak we did not attempt to identify non-smooth muscle cell types infected by this virus. Quantification of these results are shown in Figure 4B.

5) The experiments showing unstable breathing in vivo produced by injecting a thromboxane A2 receptor agonist vasoconstrictor (U46119) into the cNTS and ROb under conditions of mild hypercapnia (2-3% inspired CO_2_) are intriguing, but these experiments lack the proper control of U46119 injections into the cNTS and ROb under normocapnic conditions to determine if this alters blood pressure and produces breathing instabilities independent of any "gain-up" of RTN activity. It would also be of interest to know whether the authors have tested if larger instabilities occur with cNTS/ROb vasoconstriction at higher levels of hypercapnia.

It is important to note that this work was performed using isoflurane-anesthetized mice which require 2-3% CO_2_ in order to maintain respiratory activity (this level of CO_2_ is defined as the apneic threshold). Therefore, for this experiment we consider 2-3% CO_2_ optimum for detecting changes in respiratory activity in either direction. However, as suggested by the reviewer we did revisit these data to determine if injections of U46619 into the cNTS and ROb affected the CO_2_ apneic threshold. Consistent with the possibility that vasoconstriction increases tissue CO_2_/H^+^ levels and chemoreceptor drive, we found that in injections of U46619 into the cNTS and ROb lowered the CO_2_/H^+^ apneic threshold. These new results have been added as a new Figure 3F.

We have not tested whether this experimental manipulation results in larger instabilities at higher levels of CO_2_. This is something we plan to explore in the future in control and P2Y_2_ cKO mice.